# Universal fragment descriptors for predicting properties of inorganic crystals

Olexandr Isayev[1], Corey Oses[2], Cormac Toher[2], Eric Gossett[2], Stefano Curtarolo[2,3] & Alexander Tropsha[1]

Although historically materials discovery has been driven by a laborious trial-and-error process, knowledge-driven materials design can now be enabled by the rational combination of Machine Learning methods and materials databases. Here, data from the AFLOW repository for *ab initio* calculations is combined with Quantitative Materials Structure-Property Relationship models to predict important properties: metal/insulator classification, band gap energy, bulk/shear moduli, Debye temperature and heat capacities. The prediction's accuracy compares well with the quality of the training data for virtually any stoichiometric inorganic crystalline material, reciprocating the available thermomechanical experimental data. The universality of the approach is attributed to the construction of the descriptors: Property-Labelled Materials Fragments. The representations require only minimal structural input allowing straightforward implementations of simple heuristic design rules.

[1] Laboratory for Molecular Modeling, Division of Chemical Biology and Medicinal Chemistry, UNC Eshelman School of Pharmacy, University of North Carolina, Chapel Hill, North Carolina 27599, USA. [2] Center for Materials Genomics, Duke University, Durham, North Carolina 27708, USA. [3] Materials Science, Electrical Engineering, Physics and Chemistry, Duke University, Durham, North Carolina 27708, USA. Correspondence and requests for materials should be addressed to O.I. (email: olexandr@olexandrisayev.com) or to S.C. (email: stefano@duke.edu).

Advances in materials science are often slow and fortuitous[1]. Coupling the field's combinatorial challenges with the demanding efforts required for materials characterization makes progress uniquely difficult. The number of materials currently characterized, either experimentally[2,3] or computationally[4], pales in comparison with the anticipated potential diversity. Only considering naturally occurring elements, 9,000 crystal structure prototypes[2,3], and stoichiometric compositions, there are roughly $3 \times 10^{11}$ potential quaternary compounds and $10^{13}$ quinary combinations. Indeed, it has been estimated that the total number of theoretical materials can be as large as $10^{100}$ (ref. 5). Exacerbating the issue, standard materials characterization practices, such as calculating the band structure, can become expensive when considering finite-size scaling, charge corrections[6], and going beyond standard density functional theory (DFT) with Green's function methods such as the fully self-consistent GW approximation[7,8]. Ultimately, brute force exploration of this search space, even in high-throughput fashion[1,9], is entirely impractical.

To circumvent the issue, many knowledge-based structure–property relationships have been conjectured over the years to aid in the search for novel functional materials–ranging from the simplest empirical relationships[10] to complex advanced models[11–17]. For instance, many (semi-)empirical rules have been developed to predict band gap energies, such as those incorporating (optical[18]) electronegativity[19]. More sophisticated Machine Learning (ML) models were also developed for chalcopyrite semiconductors[20], perovskites[21], and binary compounds[22]. Unfortunately, many of these models are limited to a single family of materials, with narrow applicability outside of their training scope.

The development of such structure–property relationships has become an integral practice in the drug industry, which faces a similar combinatorial challenge. The number of potential organic molecules is estimated to be anywhere between $10^{13}$ and $10^{180}$ (ref. 23). In computational medicinal chemistry, Quantitative Structure-Activity Relationship modelling coupled with virtual screening of chemical libraries have been largely successful in the discovery of novel bioactive compounds[24].

Here, we introduce fragment descriptors of materials structure. The combination of these descriptors with ML approaches affords the development of universal models capable of accurate prediction of properties for virtually any stoichiometric inorganic crystalline material. First, the algorithm for descriptor generation is described, along with implementation of ML methods for Quantitative Materials Structure-Property Relationship (QMSPR) modelling. Next, the effectiveness of this approach is assessed through prediction of eight critical electronic and thermomechanical properties of materials, including the metal/insulator classification, band gap energy, bulk and shear moduli, Debye temperature, heat capacities (at constant pressure and volume) and thermal expansion coefficient. The impact and interaction among the most significant descriptors as determined by the ML algorithms are highlighted. As a proof-of-concept, the QMSPR models are then employed to predict thermomechanical properties for compounds previously uncharacterized, and the predictions are validated via the AEL–AGL integrated framework (Automatic Elasticity Library-Automatic GIBBS Library)[25,26]. Such predictions are of particular value as proper calculation pathways for thermomechanical properties in the most efficient scenarios still require analysis of multiple DFT-runs, elevating the cost of already expensive calculations. Finally, ML-predictions and calculations are both compared to experimental values which ultimately corroborate the validity of the approach.

Other investigations have predicted a subset of the target properties discussed here by building ML approaches where computationally obtained quantities, such as the cohesive energy, formation energy and energy above the convex hull, are part of the input data[27]. The approach presented here is orthogonal. Once trained, our proposed models achieve comparable accuracies without the need of further *ab initio* data. All necessary input properties are either tabulated or derived directly from the geometrical structures. There are advantages: (i) *a priori*, after the training, no further calculations need to be performed, (ii) *a posteriori*, the modelling framework becomes independent of the source or nature of the training data, for example, calculated versus experimental. The latter allows for rapid extension of predictions to online applications—given the geometry of a cell and the species involved, eight ML predicted properties are returned (aflow.org/aflow-ml).

## Results

**Universal property-labelled materials fragments.** Many cheminformatics investigations have demonstrated the critical importance of molecular descriptors, which are known to influence model accuracy more than the choice of the ML algorithm[28,29]. For the purposes of this investigation, fragment descriptors typically used for organic molecules were adapted to serve for materials characterization[30]. Molecular systems can be described as graphs whose vertices correspond to atoms and edges to chemical bonds. In this representation, fragment descriptors characterize subgraphs of the full 3D molecular network. Any molecular graph invariant can be uniquely represented as a linear combination of fragment descriptors. They offer several advantages over other types of chemical descriptors[31], including simplicity of calculation, storage and interpretation[32]. However, they also come with a few disadvantages. Models built with fragment descriptors perform poorly when presented with new fragments for which they were not trained. In addition, typical fragments are constructed solely with information of the individual atomic symbols (for example, C, N, Na). Such a limited context would be insufficient for modelling the complex chemical interactions within materials.

Mindful of these constraints, fragment descriptors for materials were conceptualized by differentiating atoms not by their symbols but by a plethora of well-tabulated chemical and physical properties[33]. Descriptor features comprise of various combinations of these atomic properties. From this perspective, materials can be thought of as 'coloured' graphs, with vertices decorated according to the nature of the atoms they represent[34]. Partitions of these graphs form Property-Labelled Materials Fragments (PLMF).

Figure 1 shows the scheme for constructing PLMFs. Given a crystal structure, the first step is to determine the atomic connectivity within it. In general, atomic connectivity is not a trivial property to determine within materials. Not only must the potential bonding distances among atoms be considered, but also whether the topology of nearby atoms allows for bonding. Therefore, a computational geometry approach is employed to partition the crystal structure (Fig. 1a) into atom-centred Voronoi-Dirichlet polyhedra[35,36] (Fig. 1b). This partitioning scheme was found to be invaluable in the topological analysis of metal organic frameworks, molecules, and inorganic crystals[37]. Connectivity between atoms is established by satisfying two criteria: (i) the atoms must share a Voronoi face (perpendicular bisector between neighbouring atoms), and (ii) the interatomic distance must be shorter than the sum of the Cordero covalent radii[38] to within a 0.25 Å tolerance. Here, only strong interatomic interactions are modelled, such as covalent, ionic, and metallic bonding, ignoring van der Waals interactions. Owing to the ambiguity within materials, the bond order (single/double/triple bond classification) is not considered. Taken together, the Voronoi centres that share a Voronoi face and are within the sum of their covalent radii form a three-dimensional graph defining the connectivity within the material.

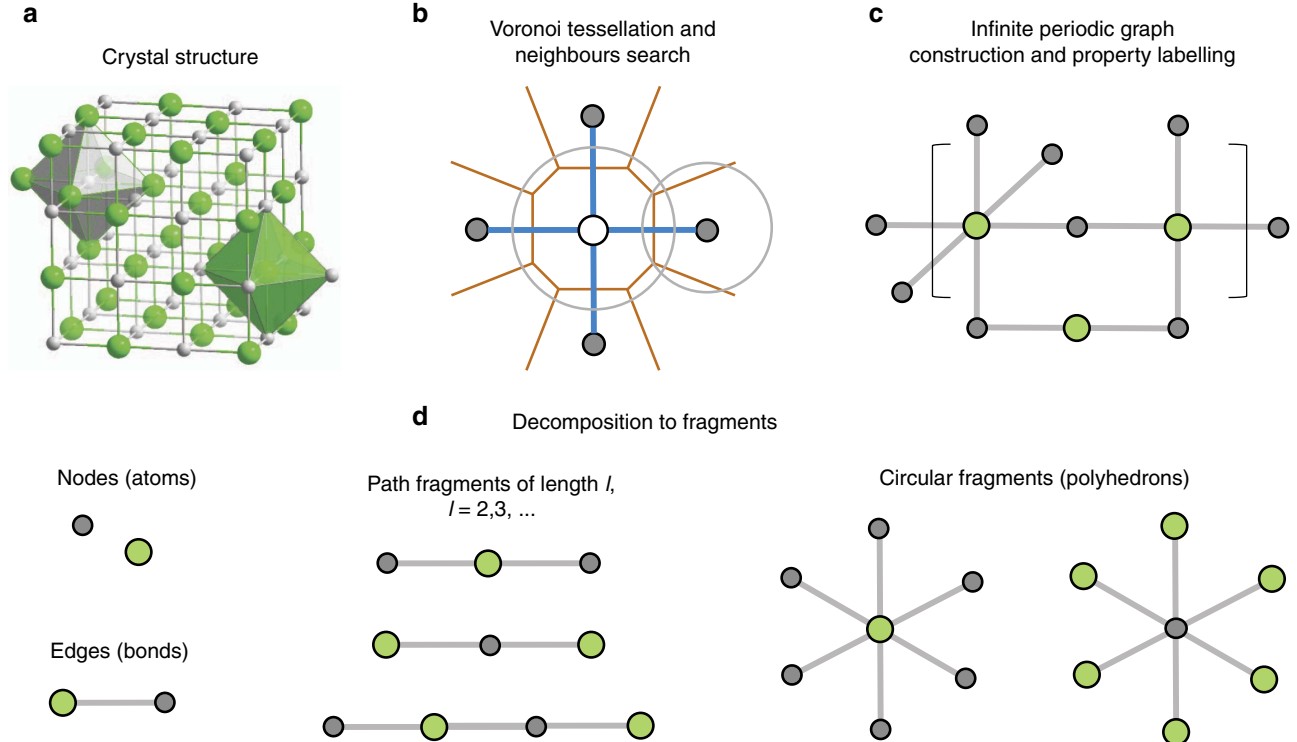

**Figure 1 | Schematic representing the construction of the Property-Labelled Materials Fragments (PLMF).** The crystal structure (**a**) is analysed for atomic neighbours via Voronoi tessellation (**b**). After property labelling, the resulting periodic graph (**c**) is decomposed into simple subgraphs (**d**).

In the final steps of the PLMF construction, the full graph and corresponding adjacency matrix (Fig. 1c) are constructed from the total list of connections. The adjacency matrix **A** of a simple graph (material) with $n$ vertices (atoms) is a square matrix ($n \times n$) with entries $a_{ij} = 1$ if atom $i$ is connected to atom $j$, and $a_{ij} = 0$ otherwise. This adjacency matrix reflects the global topology for a given system, including interatomic bonds and contacts within the crystal. The full graph is partitioned into smaller subgraphs, corresponding to individual fragments (Fig. 1d). Although there are several subgraphs to consider in general, the length $l$ is restricted to a maximum of three, where $l$ is the largest number of consecutive, non-repetitive edges in the subgraph. This restriction serves to curb the complexity of the final descriptor vector. In particular, there are two types of fragments. Path fragments are subgraphs of at most $l = 3$ that encode any linear strand of up to four atoms. Only the shortest paths between atoms are considered. Circular fragments are subgraphs of $l = 2$ that encode the first shell of nearest neighbour atoms. In this context, circular fragments represent coordination polyhedra, or clusters of atoms with anion/cation centres each surrounded by a set of its respective counter ion. Coordination polyhedra are used extensively in crystallography and mineralogy[39].

The PLMFs are differentiated by local (standard atomic/ elemental) reference properties[33], which include: (i) general properties: the Mendeleev group and period numbers ($g_P$, $p_P$), number of valence electrons ($N_V$); (ii) measured properties[33]: atomic mass ($m_{atom}$), electron affinity ($EA$), thermal conductivity ($\lambda$), heat capacity ($C$), enthalpies of atomization ($\Delta H_{at}$), fusion ($\Delta H_{fusion}$) and vaporization ($\Delta H_{vapor}$), first three ionization potentials ($IP_{1,2,3}$); and (iii) derived properties: effective atomic charge ($Z_{eff}$), molar volume ($V_{molar}$), chemical hardness ($\eta$)[33,40], covalent ($r_{cov}$)[38], absolute[41], and van der Waals radii[33], electronegativity ($\chi$) and polarizability ($\alpha_P$). Pairs of properties are included in the form of their multiplication and ratio, as

well as the property value divided by the atomic connectivity (number of neighbours in the adjacency matrix). For every property scheme **q**, the following quantities are also considered: minimum (min(**q**)), maximum (max(**q**)), total sum ($\sum$**q**), average (avg(**q**)) and standard deviation (std(**q**)) of **q** among the atoms in the material.

To incorporate information about shape, size and symmetry of the crystal unit cell, the following crystal-wide properties are incorporated: lattice parameters ($a$, $b$, $c$), their ratios ($a/b$, $b/c$, $a/c$), angles ($\alpha$, $\beta$, $\gamma$), density, volume, volume per atom, number of atoms, number of species (atom types), lattice type, point group and space group.

All aforementioned descriptors (fragment-based and crystal-wide) can be concatenated together to represent each material uniquely. After filtering out low variance ($< 0.001$) and highly correlated ($r^2 > 0.95$) features, the final feature vector captures 2,494 total descriptors.

Descriptor construction is inspired by the topological charge indices[42] and the Kier-Hall electro-topological state indices[43]. Let **M** be the matrix obtained by multiplying the adjacency matrix **A** by the reciprocal square distance matrix **D** ($D_{ij} = 1/r_{i,j}^2$):

$$\mathbf{M} = \mathbf{A} \cdot \mathbf{D}. \qquad (1)$$

The matrix **M**, called the Galvez matrix, is a square $n \times n$ matrix, where $n$ is the number of atoms in the unit cell. From **M**, descriptors of reference property **q** are calculated as

$$T^{E} = \sum_{i=1}^{n-1} \sum_{j=i+1}^{n} |q_i - q_j| M_{ij} \qquad (2)$$

and

$$T_{bond}^{E} = \sum_{\{i,j\} \in bonds} |q_i - q_j| M_{ij}, \qquad (3)$$

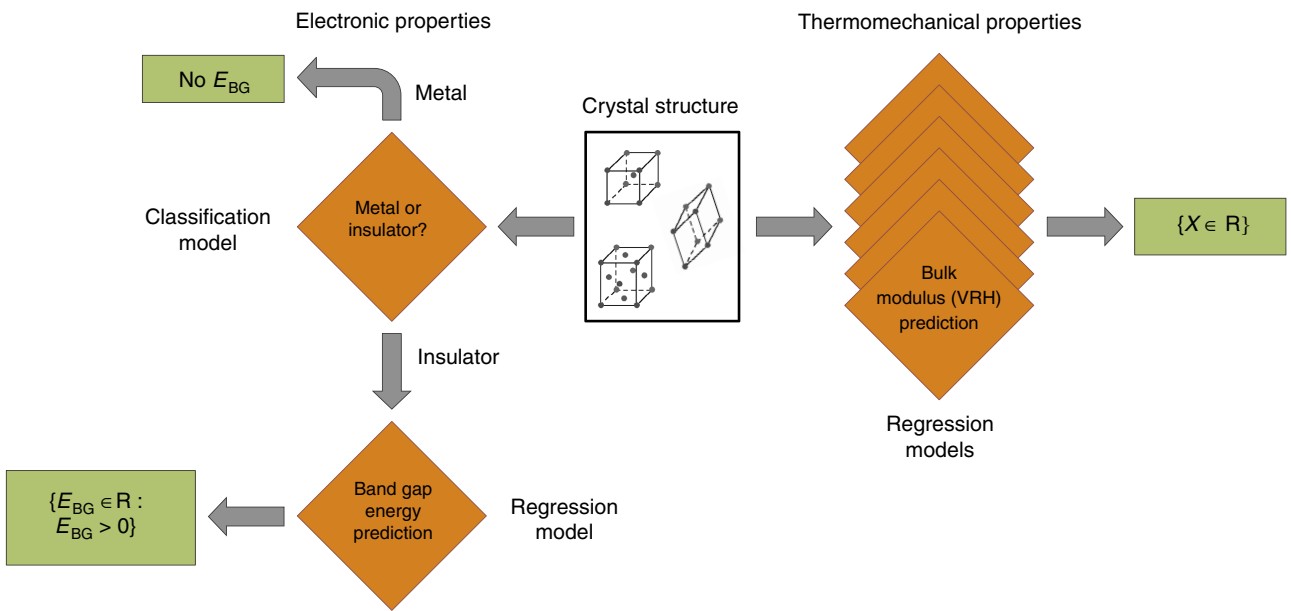

**Figure 2 | Outline of the modelling work-flow.** ML models are represented by orange diamonds. Target properties predicted by these models are highlighted in green.

where the first set of indices count over all pairs of atoms and the second is restricted to all pairs $i, j$ of bonded atoms.

**Quantitative materials structure–property relationship modelling.** In training the models, the same ML method and descriptors are employed without any hand tuning or variable selection. Specifically, models are constructed using the gradient boosting decision tree (GBDT) technique[44]. All models were validated through $y$-randomization (label scrambling). Five-fold cross validation is used to assess how well each model will generalize to an independent data set. Hyperparameters are determined with grid searches on the training set and 10-fold cross validation.

The GBDT method[44] evolved from the application of boosting methods[45] to regression trees[46]. The boosting method is based on the observation that finding many weakly accurate prediction rules can be a lot easier than finding a single, highly accurate rule[47]. The boosting algorithm calls this 'weak' learner repeatedly, at each stage feeding it a different subset of the training examples. Each time it is called, the weak learner generates a new weak prediction rule. After many iterations, the boosting algorithm combines these weak rules into a single prediction rule aiming to be much more accurate than any single weak rule.

The GBDT approach is an additive model of the following form:

$$F\big(\mathbf{x}\{\gamma_m, \mathbf{a}_m\}_1^M\big) = \sum_{m=1}^{M} \gamma_m h(\mathbf{x}; \mathbf{a}_m), \qquad (4)$$

where $h(\mathbf{x}; \mathbf{a}_m)$ are the weak learners (decision trees in this case) characterized by parameters $\mathbf{a}_m$, and $M$ is the total count of decision trees obtained through boosting.

It builds the additive model in a forward stage-wise fashion:

$$F_m(\mathbf{x}) = F_{m-1}(\mathbf{x}) + \gamma_m h(\mathbf{x}; \mathbf{a}_m). \qquad (5)$$

At each stage ($m = 1, 2, \ldots, M$), $\gamma_m$ and $\mathbf{a}_m$ are chosen to minimize the loss function $f_L$ given the current model $F_{m-1}(\mathbf{x}_i)$ for all data points (count $N$),

$$(\gamma_m, \mathbf{a}_m) = \arg\min_{\gamma, \mathbf{a}} \sum_{i=1}^{N} f_L[y_i, F_{m-1}(\mathbf{x}_i) + \gamma h(\mathbf{x}_i; \mathbf{a})]. \qquad (6)$$

Gradient boosting attempts to solve this minimization problem numerically via steepest descent. The steepest descent direction is the negative gradient of the loss function evaluated at the current model $F_{m-1}$, where the step length is chosen using line search.

An important practical task is to quantify variable importance. Feature selection in decision tree ensembles cannot differentiate between primary effects and effects caused by interactions between variables. Therefore, unlike regression coefficients, a direct comparison of captured effects is prohibited. For this purpose, variable influence is quantified in the following way[44]. Let us define the influence of variable $j$ in a single tree $h$. Consider that the tree has $l$ splits and therefore $l-1$ levels. This gives rise to the definition of the variable influence,

$$K_j^2(h) = \sum_{i=1}^{l-1} I_i^2 \mathbb{1}(x_i = j), \qquad (7)$$

where $I_i^2$ is the empirical squared improvement resulting from this split, and $\mathbb{1}$ is the indicator function. Here, $\mathbb{1}$ has a value of one if the split at node $x_i$ is on variable $j$, and zero otherwise, that is, it measures the number of times a variable $j$ is selected for splitting. To obtain the overall influence of variable $j$ in the ensemble of decision trees (count $M$), it is averaged over all trees,

$$K_j^2 = M^{-1} \sum_{m=1}^{M} K_j^2(h_m). \qquad (8)$$

The influences $K_j^2$ are normalized so that they add to one. Influences capture the importance of the variable, but not the direction of the response (positive or negative).

**Integrated modelling work-flow.** Eight predictive models are developed in this work, including: a binary classification model that predicts if a material is a metal or an insulator and seven regression models that predict: the band gap energy ($E_{BG}$) for insulators, bulk modulus ($B_{VRH}$), shear modulus ($G_{VRH}$), Debye temperature ($\theta_D$), heat capacity at constant pressure ($C_p$), heat capacity at constant volume ($C_V$), and thermal expansion coefficient ($\alpha_V$).

Figure 2 shows the overall application work-flow. A novel candidate material is first classified as a metal or an insulator. If the material is classified as an insulator, $E_{BG}$ is predicted, whereas

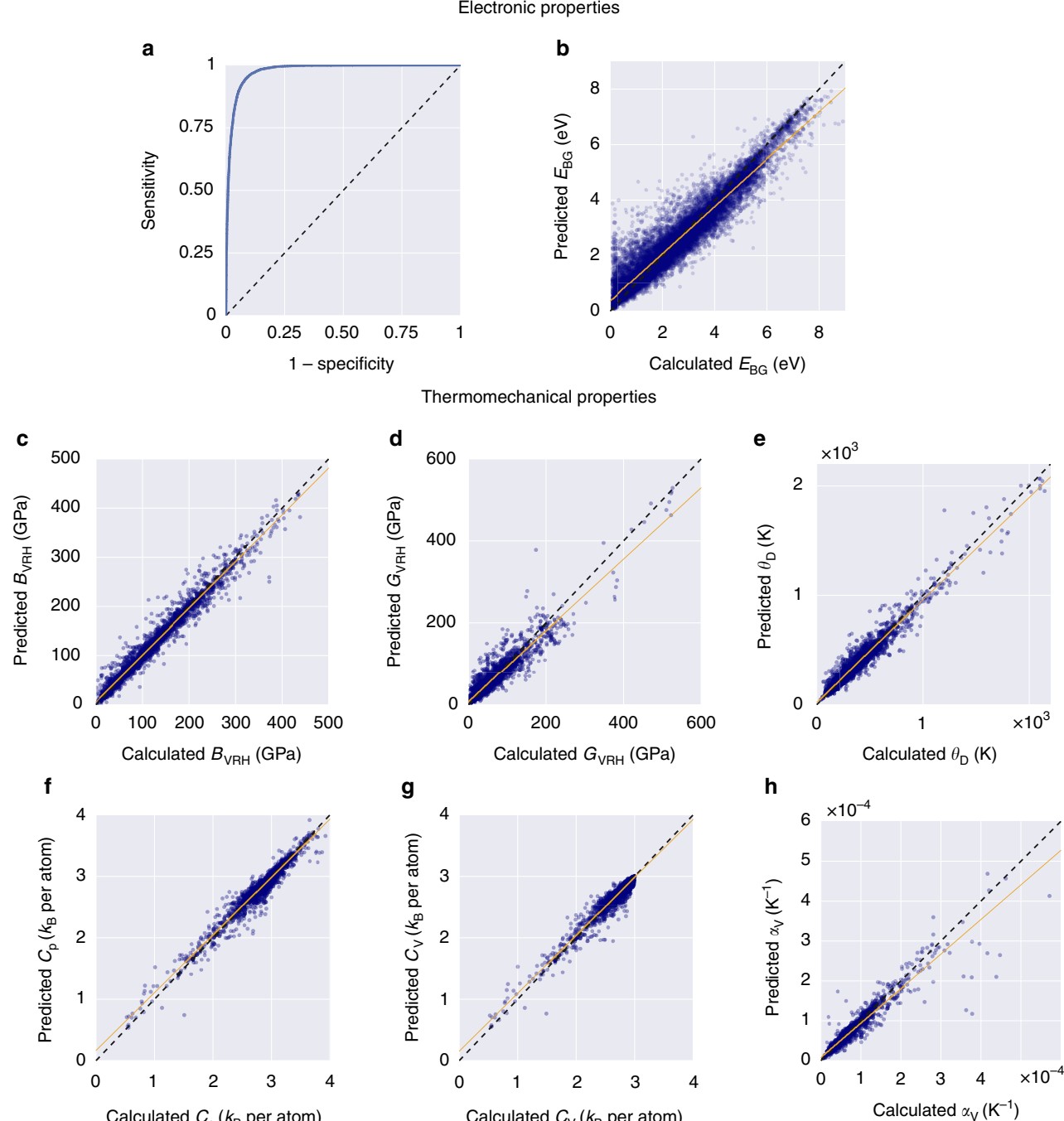

**Figure 3 | Five-fold cross validation plots for the eight ML models predicting electronic and thermomechanical properties.** (**a**) Receiver operating characteristic (ROC) curve for the classification ML model. (**b**–**h**) Predicted versus calculated values for the regression ML models: (**b**) band gap energy ($E_{BG}$), (**c**) bulk modulus ($B_{VRH}$), (**d**) shear modulus ($G_{VRH}$), (**e**) Debye temperature ($\theta_D$), (**f**) heat capacity at constant pressure ($C_P$), (**g**) heat capacity at constant volume ($C_V$) and (**h**) thermal expansion coefficient ($\alpha_V$).

classification as a metal implies that the material has no $E_{BG}$. The six thermomechanical properties are then predicted independent of the material's metal/insulator classification. The integrated modelling work-flow has been implemented as a web application at aflow.org/aflow-ml, requiring only the atomic species and positions as input for predictions.

Although all three models were trained independently, the accuracy of the $E_{BG}$ regression model is inherently dependent on the accuracy of the metal/insulator classification model in this work-flow. However, the high accuracy of the metal/insulator classification model suggests this not to be a practical concern.

**Model generalizability**. One technique for assessing model quality is fivefold cross validation, which gauges how well the model is expected to generalize to an independent data set. For each model, the scheme involves randomly partitioning the set into five groups and predicting the value of each material in one subset while training the model on the other four subsets. Hence, each subset has the opportunity to play the role of the 'test set'. Furthermore, any observed deviations in the predictions are addressed. For further analysis, all predicted and calculated results are available in Supplementary Note 2.

**Table 1 | Statistical summary of the fivefold cross validated predictions for the seven regression models.**

| Property | RMSE | MAE | $r^2$ |
|---|---|---|---|
| $E_{BG}$ | 0.51 eV | 0.35 eV | 0.90 |
| $B_{VRH}$ | 14.25 GPa | 8.68 GPa | 0.97 |
| $G_{VRH}$ | 18.43 GPa | 10.62 GPa | 0.88 |
| $\theta_D$ | 56.97 K | 35.86 K | 0.95 |
| $C_p$ | 0.09 $k_B$ per atom | 0.05 $k_B$ per atom | 0.95 |
| $C_V$ | 0.07 $k_B$ per atom | 0.04 $k_B$ per atom | 0.95 |
| $\alpha_V$ | $1.47 \times 10^{-5}$ K$^{-1}$ | $5.69 \times 10^{-6}$ K$^{-1}$ | 0.91 |

The summary corresponds with Fig. 3.

The accuracy of the metal/insulator classifier is reported as the area under the curve (AUC) of the receiver operating characteristic (ROC) plot (Fig. 3a). The ROC curve illustrates the model's ability to differentiate between metallic and insulating input materials. It plots the prediction rate for insulators (correctly versus incorrectly predicted) throughout the full spectrum of possible prediction thresholds. An area of 1.0 represents a perfect test, whereas an area of 0.5 characterizes a random guess (the dashed line). The model shows excellent external predictive power with the area under the curve at 0.98, an insulator-prediction success rate (sensitivity) of 0.95, a metal-prediction success rate (specificity) of 0.92, and an overall classification rate of 0.93. For the complete set of 26,674 materials, this corresponds to 2,103 misclassified materials, including 1,359 misclassified metals and 744 misclassified insulators. Evidently, the model exhibits positive bias toward predicting insulators, where bias refers to whether a ML model tends to over- or under-estimate the predicted property. This low false-metal rate is fortunate as the model is unlikely to misclassify a novel, potentially interesting semiconductor as a metal. Overall, the metal classification model is robust enough to handle the full complexity of the periodic table.

The results of the fivefold cross validation analysis for the band gap energy ($E_{BG}$) regression model are plotted in Fig. 3b. In addition, a statistical profile of these predictions, along with that of the six thermomechanical regression models, is provided in Table 1, which includes metrics such as the root-mean-square error (RMSE), mean absolute error (MAE), and coefficient of determination ($r^2$). Similar to the classification model, the $E_{BG}$ model exhibits a positive predictive bias. The biggest errors come from materials with narrow band gaps, that is, the scatter in the lower left corner in Fig. 3b. These materials predominantly include complex fluorides and nitrides. $N_2H_6Cl_2$ (ICSD #23145) exhibits the worst prediction accuracy with signed error SE = 3.78 eV[48]. The most underestimated materials are HCN (ICSD #76419) and $N_2H_6Cl_2$ (ICSD #240903) with SE = − 2.67 and − 3.19 eV[49,50], respectively. This is not surprising considering that all three are molecular crystals. Such systems are anomalies in the ICSD, and fit better in other databases, such as the Cambridge Structural Database[51]. Overall, 10,762 materials are predicted within 25% accuracy of calculated values, whereas 824 systems have errors over 1 eV.

Figure 3c–h and Table 1 showcase the results of the fivefold cross validation analysis for the six thermomechanical regression models. For both bulk ($B_{VRH}$) and shear ($G_{VRH}$) moduli, over 85% of materials are predicted within 20 GPa of their calculated values. The remaining models also demonstrate high accuracy, with at least 90% of the full training set ($>2,546$ systems) predicted to within 25% of the calculated values. Significant outliers in predictions of the bulk modulus include graphite (ICSD #187640, SE = 100 GPa, likely due to extreme anisotropy) and two

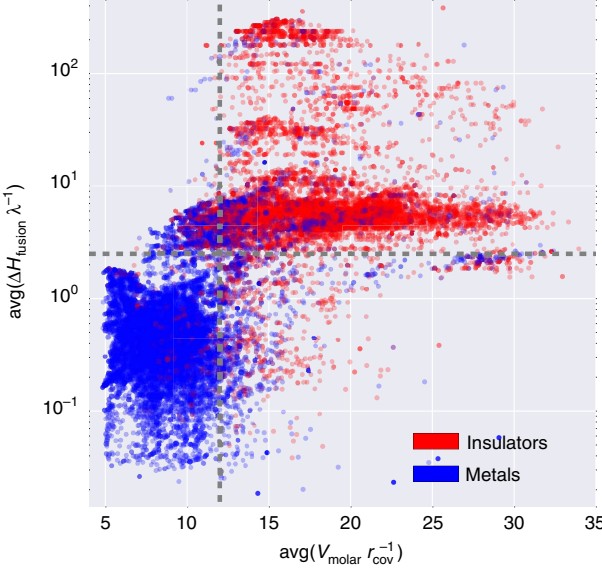

**Figure 4 | Semi-log scatter plot of the full data set (26,674 unique materials) in a dual-descriptor space.** avg($\Delta H_{fusion}\lambda^{-1}$) versus avg($V_{molar}r_{cov}^{-1}$). Insulators and metals are coloured in red and blue, respectively.

theoretical high-pressure boron nitrides (ICSD #162873 and #162874, under-predicted by over 110 GPa)[52,53]. Other theoretical systems are ill-predicted throughout the six properties, including ZN (ICSD #161885), $CN_2$ (ICSD #247676), $C_3N_4$ (ICSD #151782) and CH (ICSD #187642)[52,54–56]. Predictions for the $G_{VRH}$, Debye temperature ($\theta_D$), and thermal expansion coefficient ($\alpha_V$) tend to be slightly underestimated, particularly for higher calculated values. In addition, mild scattering can be seen for $\theta_D$ and $\theta_V$, but not enough to have a significant impact on the error or correlation metrics.

Despite minimal deviations, both RMSE and mean absolute error are within 4% of the ranges covered for each property, and the predictions demonstrate excellent correlation with the calculated properties. Note the tight clustering of points just below $3 k_B$ per atom for the heat capacity at constant volume ($C_V$). This is due to $C_V$ saturation in accordance with the Dulong-Petit law occurring at or below 300 K for many compounds.

**Model interpretation.** Model interpretation is of paramount importance in any ML study. The significance of each descriptor is determined in order to gain insight into structural features that impact molecular properties of interest. Interpretability is a strong advantage of decision tree methods, particularly with the GBDT approach. One can quantify the predictive power of a specific descriptor by analysing the reduction of the RMSE at each node of the tree.

Partial dependence plots offer yet another opportunity for GBDT model interpretation. Similar to the descriptor significance analysis, partial dependence resolves the effect of a variable (descriptor) on a property, but only after marginalising over all other explanatory variables[57]. The effect is quantified by the change of that property as relevant descriptors are varied. The plots themselves highlight the most important interactions among relevant descriptors as well as between properties and their corresponding descriptors. Although only the most important descriptors are highlighted and discussed, an exhaustive list of relevant descriptors and their relative contributions can be found in Supplementary Note 1.

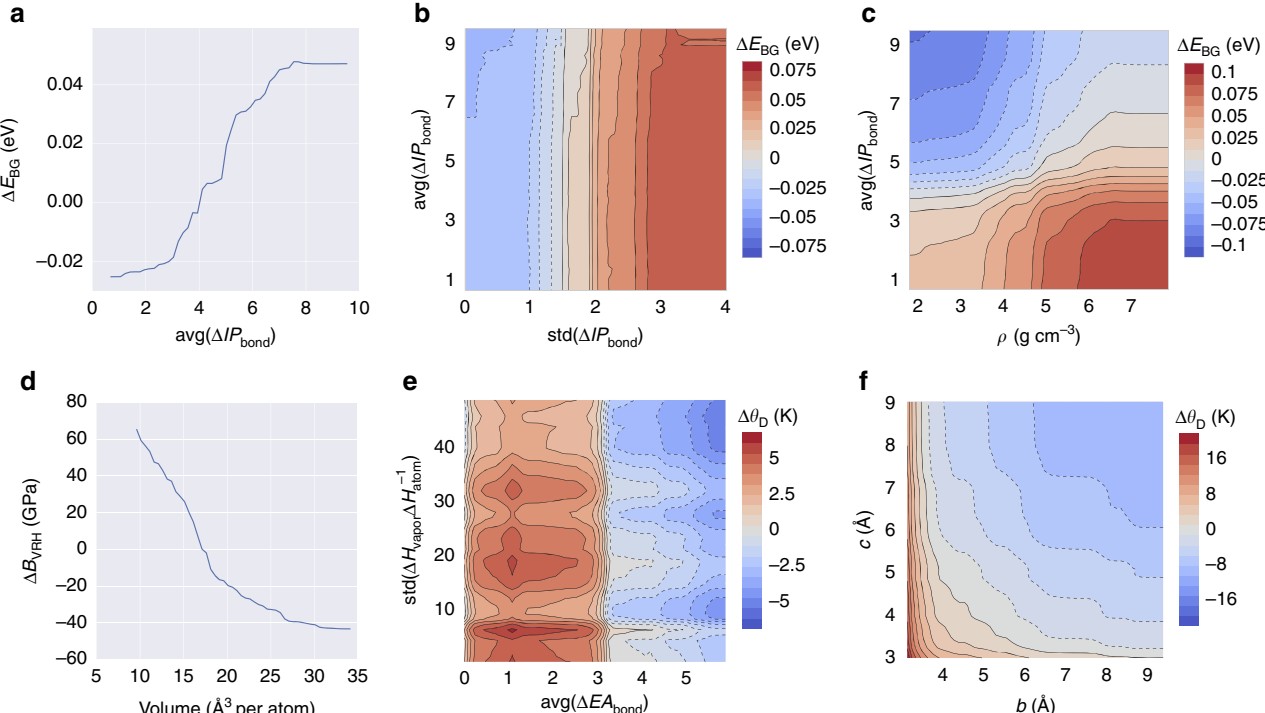

**Figure 5 | Partial dependence plots of the $E_{BG}$, $B_{VRH}$ and $\theta_D$ models.** (**a**) Partial dependence of $E_{BG}$ on the avg($\Delta IP_{bond}$) descriptor. For $E_{BG}$, the 2D interaction between std($\Delta IP_{bond}$) and avg($\Delta IP_{bond}$) and between $\rho$ (density) and avg($\Delta IP_{bond}$) are illustrated in panels (**b,c**), respectively. (**d**) Partial dependence of the $B_{VRH}$ on the crystal volume per atom descriptor. For $\theta_D$, the 2D interaction between avg($\Delta EA_{bond}$) and std($\Delta H_{vapor}\Delta H_{atom}^{-1}$) and between crystal lattice parameters $b$ and $c$ are illustrated in panels (**e,f**), respectively.

For the metal/insulator classification model, the descriptor significance analysis shows that two descriptors have the highest importance (equally), namely avg($\Delta H_{fusion}\lambda^{-1}$) and avg($V_{molar}r_{cov}^{-1}$). avg($\Delta H_{fusion}\lambda^{-1}$) is the ratio between the fusion enthalpy ($\Delta H_{fusion}$) and the thermal conductivity ($\lambda$) averaged over all atoms in the material, and avg($V_{molar}r_{cov}^{-1}$) is the ratio between the molar volume ($V_{molar}$) and the covalent radius ($r_{cov}$) averaged over all atoms in the material. Both descriptors are simple node-specific features. The presence of these two prominent descriptors accounts for the high accuracy of the classification model.

Figure 4 shows the projection of the full dataset onto the dual-descriptor space of avg($\Delta H_{fusion}\lambda^{-1}$) and avg($V_{molar}r_{cov}^{-1}$). In this 2D space, metals and insulators are substantially partitioned. To further resolve this separation, the plot is split into four quadrants (see dashed lines) with an origin approximately at avg($V_{molar}r_{cov}^{-1}$)=11, avg($\Delta H_{fusion}\lambda^{-1}$)=2. Insulators are predominately located in quadrant I. There are several clusters (one large and several small) parallel to the $x$ axis. Metals occupy a compact square block in quadrant III within intervals $5<$avg($V_{molar}r_{cov}^{-1}$)$<12$ and $0.02<$avg($\Delta H_{fusion}\lambda^{-1}$)$<2$. Quadrant II is mostly empty with a few materials scattered about the origin. In the remaining quadrant (IV), materials have mixed character.

Analysis of the projection shown in Fig. 4 suggests a simple heuristic rule: all materials within quadrant I are classified as insulators ($E_{BG}>0$), and all materials outside of this quadrant are metals. Remarkably, this unsupervised projection approach achieves a very high classification accuracy of 86% for the entire dataset of 26,674 materials. The model misclassifies only 3,621 materials: 2,414 are incorrectly predicted as insulators and 1,207 are incorrectly predicted as metals. This example illustrates how careful model analysis of the most significant descriptors can yield simple heuristic rules for materials design.

The regression model for the band gap energy ($E_{BG}$) is more complex. There are a number of descriptors in the model with comparable contributions, and thus, all individual contributions are small. This is expected as a number of conditions can affect $E_{BG}$. The most important are avg($\chi Z_{eff}^{-1}$) and avg($C\lambda^{-1}$) with significance scores of 0.075 and 0.071, respectively, where $\chi$ is the electronegativity, $Z_{eff}$ is the effective nuclear charge, $C$ is the specific heat capacity and $\lambda$ is the thermal conductivity of each atom.

Figure 5 shows partial dependence plots focusing on ($\Delta IP_{bond}$) as an example. It is derived from edge fragments of bonded atoms ($l=1$) and defined as an absolute difference in ionization potentials averaged over the material. In other words, it is a measure of bond polarity, similar to electronegativity. Figure 5a shows a steady monotonic increase in $E_{BG}$ for larger values of ($\Delta IP_{bond}$). The effect is small, but captures an expected physical principle: polar inorganic materials (for example, oxides, fluorides) tend to have larger $E_{BG}$.

Given the number of significant interactions involved with this phenomenon, tailoring $E_{BG}$ involves the optimization of a highly non-convex, multidimensional object. Figure 5b illustrates a 2D slice of this object as std($\Delta IP_{bond}$) and avg($\Delta IP_{bond}$) vary simultaneously. Like avg($\Delta IP_{bond}$), std($\Delta IP_{bond}$) is the s.d. of the set of absolute differences in $IP$ among all bonded atoms. In the context of these two variables, $E_{BG}$ responds to deviations in $\Delta IP_{bond}$ among the set of bonded atoms, but remains constant across shifts in avg($\Delta IP_{bond}$). This suggests an opportunity to tune $E_{BG}$ by considering another composition that varies the deviations among bond polarities. Alternatively, a desired $E_{BG}$ can be maintained by considering another composition that preserves the deviations among bond polarities, even as the overall average shifts. Similarly, Fig. 5c shows the partial dependence on both the density ($\rho$) and avg($\Delta IP_{bond}$). Contrary

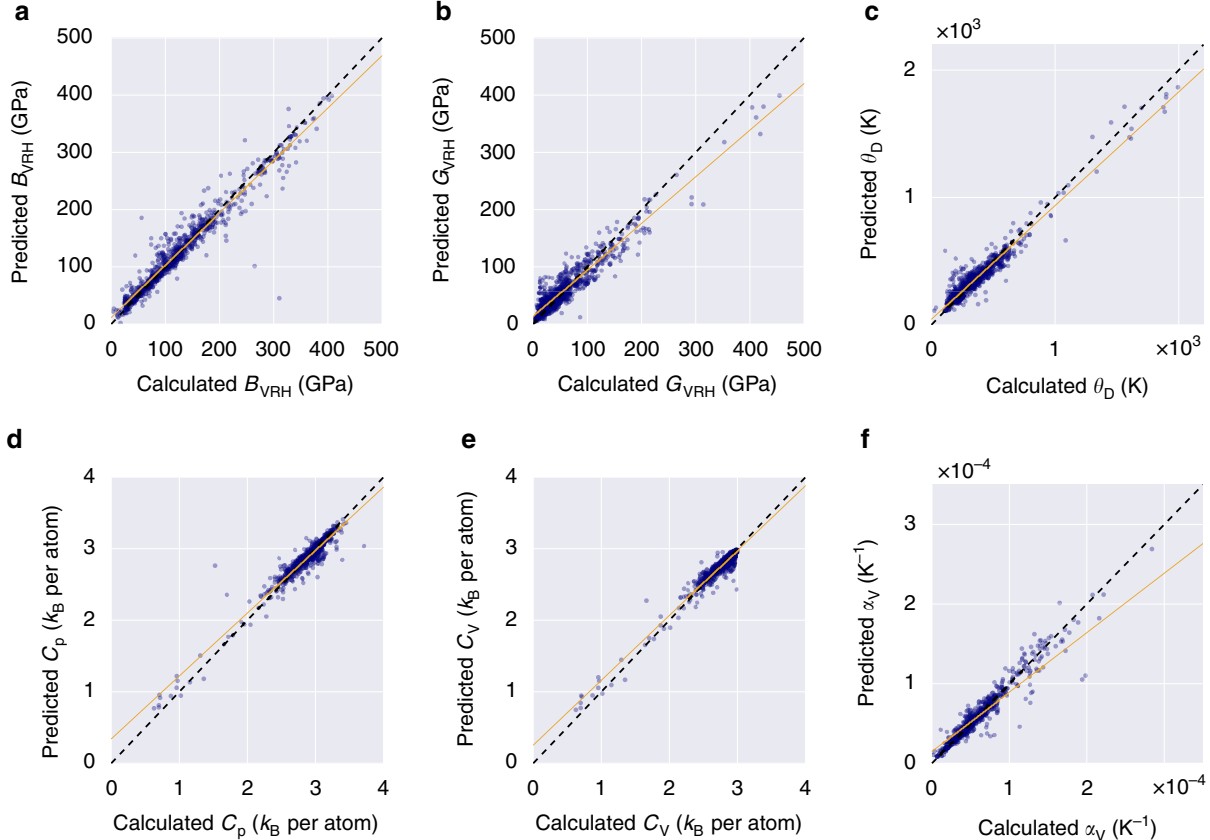

**Figure 6 | Model performance evaluation for the six ML models predicting thermomechanical properties of 770 characterized materials.** Predicted versus calculated values for the regression ML models: (**a**) bulk modulus ($B_{VRH}$), (**b**) shear modulus ($G_{VRH}$), (**c**) Debye temperature ($\theta_D$), (**d**) heat capacity at constant pressure ($C_P$), (**e**) heat capacity at constant volume ($C_V$), and (**f**) thermal expansion coefficient ($\alpha_V$).

**Table 2 | Statistical summary of the predictions for the six thermomechanical regression models.**

| Property | RMSE | MAE | $r^2$ |
|---|---|---|---|
| $B_{VRH}$ | 21.13 GPa | 12.00 GPa | 0.93 |
| $G_{VRH}$ | 18.94 GPa | 13.31 GPa | 0.90 |
| $\theta_D$ | 64.04 K | 42.92 K | 0.93 |
| $C_P$ | 0.10 $k_B$ per atom | 0.06 $k_B$ per atom | 0.92 |
| $C_V$ | 0.07 $k_B$ per atom | 0.05 $k_B$ per atom | 0.95 |
| $\alpha_V$ | $1.95 \times 10^{-5} \text{K}^{-1}$ | $5.77 \times 10^{-6} \text{ K}^{-1}$ | 0.76 |

The summary corresponds with Fig. 6.

to the previous trend, larger avg($\Delta IP_{bond}$) values correlate with smaller $E_{BG}$, particularly for low density structures. Materials with higher density and lower avg($\Delta IP_{bond}$) tend to have higher $E_{BG}$. Considering the elevated response (compared with Fig. 5b), the inverse correlation of $E_{BG}$ with the average bond polarity in the context of density suggests an even more effective means of tuning $E_{BG}$.

A descriptor analysis of the thermomechanical property models reveals the importance of one descriptor in particular, the volume per atom of the crystal. This conclusion certainly resonates with the nature of these properties, as they generally correlate with bond strength[26]. Figure 4d exemplifies such a relationship, which shows the partial dependence plot of the bulk modulus ($B_{VRH}$) on the volume per atom. Tightly bound atoms are generally indicative of stronger bonds. As the interatomic distance increases, properties like $B_{VRH}$ generally reduce.

Two of the more interesting dependence plots are also shown in Fig. 5e,f, both of which offer opportunities for tuning the Debye temperature ($\theta_D$). Figure 5e illustrates the interactions among two descriptors, the absolute difference in electron affinities among bonded atoms averaged over the material (avg($\Delta EA_{bond}$)), and the s.d. of the set of ratios of the enthalpies of vaporization ($\Delta H_{vapor}$) and atomization $\Delta H_{atom}$) for all atoms in the material ($\text{std}(\Delta H_{vapor}\Delta H_{atom}^{-1})$). Within these dimensions, two distinct regions emerge of increasing/ decreasing $\theta_D$ separated by a sharp division at about avg($\Delta EA_{atom}$)=3. Within these partitions, there are clusters of maximum gradient in $\theta_D$—peaks within the left partition and troughs within the right. The peaks and troughs alternate with varying $\text{std}(\Delta H_{vapor}\Delta H_{atom}^{-1})$. Although $\text{std}(\Delta H_{vapor} \Delta H_{atom}^{-1})$ is not an immediately intuitive descriptor, the alternating clusters may be a manifestation of the periodic nature of $\Delta H_{vapor}$ and $\Delta H_{atom}$ (ref. 58). As for the partitions themselves, the extremes of avg($\Delta EA_{atom}$) characterize covalent and ionic materials, as bonded atoms with similar $EA$ are likely to share electrons, whereas those with varying $EA$ prefer to donate/accept electrons. Considering that $EA$ is also periodic, various opportunities for carefully tuning $\theta_D$ should be available.

Finally, Fig. 5f shows the partial dependence of $\theta_D$ on the lattice parameters $b$ and $c$. It resolves two notable correlations: (i) uniformly increasing the cell size of the system decreases $\theta_D$, but (ii) elongating the cell ($c/b \gg 1$) increases it. Again, (i) can be attributed to the inverse relationship between volume per atom and bond strength, but does little to address (ii). Nevertheless, the connection between elongated, or layered, systems and the Debye temperature is certainly not surprising—anisotropy can be

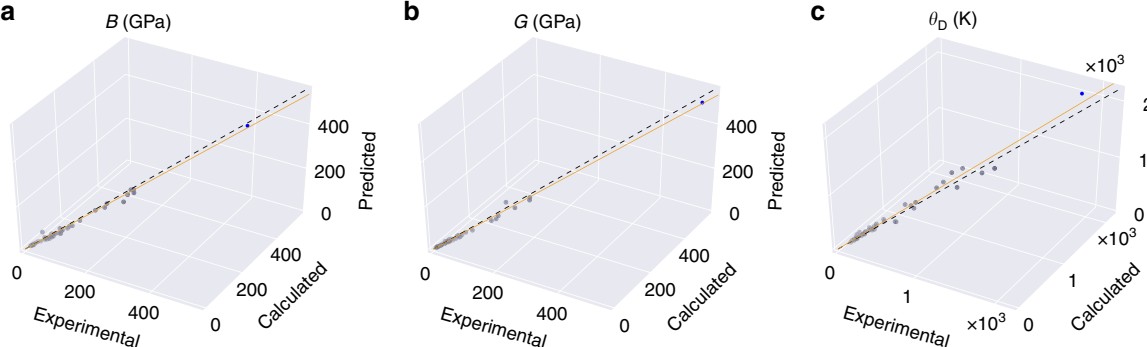

**Figure 7 | Comparison of the AEL—AGL calculations and ML predictions with experimental values for three thermomechanical properties.** (**a**) bulk modulus ($B$), (**b**) shear modulus ($G$), and (**c**) Debye temperature ($\theta_D$).

**Table 3 | Statistical summary of the AEL–AGL calculations and ML predictions versus experimental values for three thermomechanical properties.**

| Property | RMSE | | MAE | | $r^2$ | |
|---|---|---|---|---|---|---|
| | exp. versus calc. | exp. versus pred. | exp. versus calc. | exp. versus pred. | exp. versus calc. | exp. versus pred. |
| $B$ | 8.90 GPa | 10.77 GPa | 6.36 GPa | 8.12 GPa | 0.99 | 0.99 |
| $G$ | 7.29 GPa | 9.15 GPa | 4.76 GPa | 6.09 GPa | 0.99 | 0.99 |
| $\theta_D$ | 76.13 K | 65.38 K | 49.63 K | 42.92 K | 0.97 | 0.97 |

The summary corresponds with Fig. 7.

leveraged to enhance phonon-related interactions associated with thermal conductivity[59] and superconductivity[60–62]. Although the domain of interest is quite narrow, the impact is substantial, particularly in comparison with that shown in Fig. 5e.

**Model validation.** Although the expected performances of the ML models can be projected through fivefold cross validation, there is no substitute for validation against an independent dataset. The ML models for the thermomechanical properties were leveraged to make predictions for materials previously uncharacterized, and these predictions were subsequently validated via the AEL–AGL integrated framework[25,26]. Figure 6 illustrates the models' performance on the set of 770 additional materials, with relevant statistics displayed in Table 2. For further analysis, all predicted and calculated results are available in Supplementary Note 3.

Comparing with the results of the generalizability analysis shown in Fig. 3 and Table 1, the overall errors are consistent with fivefold cross validation. Five out of six models have $r^2$ of 0.9 or higher. However, the $r^2$ value for the thermal expansion coefficient ($\alpha_V$) is lower than forecasted. The presence of scattering suggests the need for a larger training set—as new, much more diverse materials were likely introduced in the test set. This is not surprising considering the number of variables that can affect thermal expansion[63]. Otherwise, the accuracy of these predictions confirm the effectiveness of the PLMF representation, which is particularly compelling considering: (i) the limited diversity training dataset (only ∼11% as large as that available for predicting the electronic properties) and (ii) the relative size of the test set (over a quarter the size of the training set).

In the case of the bulk modulus ($B_{VRH}$), 665 systems (86% of test set) are predicted within 25% of calculated values. Only the predictions of four materials, Bi (ICSD #51674), PrN (ICSD #168643), $Mg_3Sm$ (ICSD #104868), and ZrN (ICSD #161885), deviate beyond 100 GPa from calculated values. Bi is a high-

pressure phase (Bi-III) with a caged, zeolite-like structure[64]. The structures of zirconium nitride (wurtzite phase) and praseodymium nitride (B3 phase) were hypothesized and investigated via DFT calculations[54,65] and have yet to be observed experimentally.

For the shear modulus ($G_{VRH}$) 482 materials (63% of the test set) are predicted within 25% of calculated values. Just one system, $C_3N_4$ (ICSD #151781), deviates beyond 100 GPa from its calculated value. The Debye temperature ($\theta_D$) is predicted to within 50 K accuracy for 540 systems (70% of the test set). $BeF_2$ (ICSD #173557), yet another cage (sodalite) structure[66], has among the largest errors in three models including $\theta_D$ (SE = − 423 K) and both heat capacities ($C_p$: SE = 0.65 $k_B$ per atom; $C_V$: SE = 0.61 $k_B$ per atom). Similar to other ill-predicted structures, this polymorph is theoretical, and has yet to be synthesized.

**Comparison with experiments.** A comparison between calculated, predicted and experimental results is presented in Fig. 7, with relevant statistics summarized in Table 3. Data are considered for the bulk modulus $B$, shear modulus $G$, and (acoustic) Debye temperature $\theta_a$ for 45 well-characterized materials with diamond (SG# 227, AFLOW prototype A_cF8_227_a), zincblende (SG# 216, AB_cF8_216_c_a), rocksalt (SG# 225, AB_cF8_225_a_b), and wurtzite (SG# 186, AB_hP4_186_b_b) structures[67,68]. Experimental $B$ and $G$ are compared with the $B_{VRH}$ and $G_{VRH}$ values predicted here, and $\theta_a$ is converted to the traditional Debye temperature $\theta_D = \theta_a n^{1/3}$, where $n$ is the number of atoms in the unit cell. All relevant values are listed in Supplementary Note 4.

Excellent agreement is found between experimental and calculated values, but more importantly, between experimental and predicted results. With error metrics close to or under expected tolerances from the generalizability analysis, the comparison highlights effective experimental confidence in the approach. The experiments/prediction validation is clearly the ultimate objective of the research presented here.

**Discussion**
Traditional trial-and-error approaches have proven ineffective in discovering practical materials. Computational models developed with ML techniques may provide a truly rational approach to materials design. Typical high-throughput DFT screenings involve exhaustive calculations of all materials in the database, often without consideration of previously calculated results. Even at high-throughput rates, an average DFT calculation of a medium size structure (∼50 atoms per unit cell) takes ∼1,170 CPU-hours of calculations or about 37 h on a 32-CPU cores node. However, in many cases, the desired range of values for the target property is known. For instance, the optimal band gap energy and thermal conductivity for

optoelectronic applications will depend on the power and voltage conditions of the device[63,69]. Such cases offer an opportunity to leverage previous results and savvy ML models, such as those developed in this work, for rapid pre-screening of potential materials. Researchers can quickly narrow the list of candidate materials and avoid many extraneous DFT calculations—saving money, time and computational resources. This approach takes full advantage of previously calculated results, continuously accelerating materials discovery. With prediction rates of about 0.1 s per material, the same 32-CPU cores node can screen over 28 million material candidates per day with this framework.

Furthermore, interaction diagrams as depicted in Fig. 5 offer a pathway to design materials that meet certain constraints and requirements. For example, substantial differences in thermal expansion coefficients among the materials used in high-power, high-frequency optoelectronic applications leads to bending and cracking of the structure during the growth process[63,69]. Not only would this work-flow facilitate the search for semiconductors with large band gap energies, high Debye temperatures (thermal conductivity), but also materials with similar thermal expansion coefficients.

Although the models themselves demonstrate excellent predictive power with minor deviations, outlier analysis reveals theoretical structures to be among the worst offenders. This is not surprising, as the true stability conditions (for example, high-pressure/high-temperature) have yet to be determined, if they exist at all. The ICSD estimates that structures for over 7,000 materials (or roughly 4%) come from calculations rather than actual experiment. Such discoveries exemplify yet another application for ML modelling, rapid/robust curation of large data sets.

To improve large-scale high-throughput computational screening for the identification of materials with desired properties, fast and accurate data mining approaches should be incorporated into the standard work-flow. In this work, we developed a universal QMSPR framework for predicting electronic properties of inorganic materials. Its effectiveness is validated through the prediction of eight key materials properties for stoichiometric inorganic crystalline materials, including the metal/insulator classification, band gap energy, bulk and shear moduli, Debye temperature, heat capacity (at constant pressure and volume) and thermal expansion coefficient. Its applicability extends to all 230 space groups and the vast majority of elements in the periodic table. All models are freely available at aflow.org/aflow-ml.

## Methods

**Data preparation.** Two independent data sets were prepared for the creation and validation of the ML models. The training set includes electronic[4,70–74] and thermomechanical properties[25,26] for a broad diversity of compounds already characterized in the AFLOW database. This set is used to build and analyse the ML models, one model per property. The constructed thermomechanical models are then employed to make predictions of previously uncharacterized compounds in the AFLOW database. Based on these predictions and consideration of computational cost, several compounds are selected to validate the models' predictive power. These compounds and their computed properties define the test set. The compounds used in both data sets are specified in Supplementary Notes 2 and 3, respectively.

**Training set. I.** Band gap energy data for 49,934 materials were extracted from the AFLOW repository[4,70–74], representing ∼60% of the known stoichiometric inorganic crystalline materials listed in the Inorganic Crystal Structure Database (ICSD)[2,3]. Although these band gap energies are generally underestimated with respect to experimental values[75], DFT + $U$ is robust enough to differentiate between metallic (no $E_{BG}$) and insulating ($E_{BG} > 0$) systems[76]. In addition, errors in band gap energy prediction are typically systematic. Therefore, the band gap energy values can be corrected *ad hoc* with fitting schemes[77,78]. Prior to model development, both ICSD and AFLOW data were curated: duplicate entries, erroneous structures, and ill-converged calculations were corrected or removed. Noble gases crystals are not considered. The final data set consists of 26,674 unique

materials (12,862 with no $E_{BG}$ and 13,812 with $E_{BG} > 0$), covering the seven lattice systems, 230 space groups and 83 elements (H-Pu, excluding noble gases, Fr, Ra, Np, At and Po). All referenced DFT calculations were performed with the Generalized Gradient Approximation PBE[79] exchange-correlation functional and projector-augmented wavefunction potentials[80,81] according to the AFLOW Standard for High-Throughput Computing[76]. The Standard ensures reproducibility of the data, and provides visibility/reasoning for any parameters set in the calculation, such as accuracy thresholds, calculation pathways, and mesh dimensions. **II.** Thermomechanical properties data for just over 3,000 materials were extracted from the AFLOW repository[26]. These properties include the bulk modulus, shear modulus, Debye temperature, heat capacity at constant pressure, heat capacity at constant volume, and thermal expansion coefficient, and were calculated using the AEL–AGL integrated framework[25,26]. The AEL (AFLOW Elasticity Library) method[26] applies a set of independent normal and shear strains to the structure, and then fits the calculated stress tensors to obtain the elastic constants[82]. These can then be used to calculate the elastic moduli in the Voigt and Reuss approximations, as well as the Voigt-Reuss-Hill (VRH) averages that are the values of the bulk and shear moduli modelled in this work. The AGL (AFLOW GIBBS Library) method[25] fits the energies from a set of isotropically compressed and expanded volumes of a structure to a quasiharmonic Debye-Grüneisen model[83] to obtain thermomechanical properties, including the bulk modulus, Debye temperature, heat capacity and thermal expansion coefficient. AGL has been combined with AEL in a single work-flow, so that it can utilize the Poisson ratios obtained from AEL to improve the accuracy of the thermal properties predictions[26]. After a similar curation of ill-converged calculations, the final data set consists of 2,829 materials. It covers the seven lattice systems, includes unary, binary and ternary compounds, and spans broad ranges of each thermomechanical property, including high thermal conductivity systems such as C (ICSD #182729), BN (ICSD #162874), BC$_5$ (ICSD #166554), CN$_2$ (ICSD #247678), MnB$_2$ (ICSD #187733) and SiC (ICSD #164973), as well as low thermal conductivity systems such as Hg$_{33}$(Rb,K)$_3$ (ICSD #410567 and #410566), Cs$_6$Hg$_{40}$ (ICSD #240038), Ca$_{16}$Hg$_{36}$ (ICSD #107690), CrTe (ICSD #181056) and Cs (ICSD #426937). Many of these systems additionally exhibit extreme values of the bulk and shear moduli, such as C (high bulk and shear moduli) and Cs (low bulk and shear moduli). Interesting systems such as RuC (ICSD #183169) and NbC (ICSD #189090) with a high bulk modulus ($B_{VRH}$ = 317.92 GPa, 263.75 GPa) but low shear modulus ($G_{VRH}$ = 16.11 GPa, 31.86 GPa) also populate the set.

**Test set.** Although nearly all ICSD compounds are characterized electronically within the AFLOW database, most have not been characterized thermo-mechanically owing to the added computational cost. This presented an opportunity to validate the ML models. Of the remaining compounds, several were prioritized for immediate characterization via the AEL–AGL integrated framework[25,26]. In particular, focus was placed on systems predicted to have a large bulk modulus, as this property is expected to scale well with the other aforementioned thermomechanical properties[25,26]. The set also includes various other small cell, high symmetry systems expected to span the full applicability domains of the models. This effort resulted in the characterization of 770 additional compounds.

**Data availability.** All the *ab initio* data are freely available to the public as part of the AFLOW online repository and can be accessed through aflow.org following the REST-API interface[70].

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

## Acknowledgements

A.T. and O.I. acknowledge support from DOD-ONR (N00014-13-1-0028 and N00014-16-1-2311) and ITS Research Computing Center at UNC. Development of the web service was supported by the Russian Scientific Foundation (# 14-43-00024). O.I. acknowledges Extreme Science and Engineering Discovery Environment (XSEDE) award DMR-110088, which is supported by National Science Foundation grant number ACI-1053575. S.C. and C.T. acknowledge support from DOD-ONR (N00014-13-1-0030, N00014-13-1-0635), DOE (DE-AC02-05CH11231, specifically BES Grant # EDCBEE) and the Duke University Center for Materials Genomics. C.O. acknowledges support from the National Science Foundation Graduate Research Fellowship under Grant No. DGF-1106401. AFLOW calculations were performed at the Duke University Center for Materials Genomics.

## Author contributions

A.T. and S.C. designed the study. O.I. developed and implemented the method. C.O. and C.T. prepared the data and worked with the AFLOW database. E.G. developed the open-access online application available at aflow.org/aflow-ml leveraging the ML models.

O.I and C.O. contributed equally to the work. All authors discussed the results and their implications and contributed to the writing of the article.

## Additional information

**Competing interests:** The authors declare no competing financial interests.

**Publisher's note**: 

