## [Peer Review File · Nature Communications]

Reviewers' comments:

Reviewer #1 (Remarks to the Author):

The paper reports a novel graph oriented machine learning technique for analyzing the AFLOWlib repository and predicting mainly band-gaps from the connectivity and a large number of atomic descriptors.

While I like the paper, it is a very clearly written and illustrated attempt at an interesting field, I have difficulties finding arguments for why it should be published in Nat. Comm., I think it belongs in a more specialized journal like J.Chem.Phys.

My major objections are:

1. The choice of studying E-fermi is very questionable. In a PW code this is a parameter set with respect to $G=0$ and carries no physical significance. It says nothing about band alignment as otherwise used as a justification by the authors in the intro and conclusion.
2. As the authors mention, there are now a number of ML studies of structure property relationships. I don't see how the present really stands out. Especially as no new predictions about materials are made or surprising insights are offered. It is of methodological interest and belongs in a methodological journal.
3. The average predictive quality might be high, but large failures of the model are numerous. This would strongly limit a screening based on the model. Again it is disappointing that no predictions are made and validated.

Smaller point:

The "Fragment construction" subsection is a bit repetitive (compare p2c2)

Reviewer #2 (Remarks to the Author):

The main claim of this paper is that the fragment material descriptors introduced by the authors in combination with machine learning approaches allows the development of universal models capable of accurately predicting electronic properties for inorganic materials.

To the best of my knowledge these claims are novel. They are of course a continuation of earlier studies by the authors and others as cited in the paper itself. Machine learning approaches for predicting materials properties is a very active field at the forefront of research and this paper will most definitely be of interest to other researchers in this field. The ideas promoted in this paper may be expected to influence thinking in this field in two main directions at least: (i) establishing feasibility of ML methods in materials properties predictions and (ii) specific contribution of the materials descriptors employed here including the material fragments and the mix of empirical and computational properties.

The authors claim is supported by three separate analyses: (i) classification of metal/non-metal materials (ii) prediction of the Fermi energy relative to DFT (iii) prediction of the band gap relative to DFT. While the proposed method is quite successful in predicting these quantities there are some weaknesses arising from the choice of properties. In particular, the Fermi energy is a computational concept and depends strongly on the choice of model and the band gap is a notoriously difficult problem in DFT and the error relative to experiment can be tens of percent.

I think that the authors' case would be strengthened considerably by modelling a mechanical parameter, such as the bulk modulus, in addition to the purely electronic properties considered. Another issue would be to compare the predicted band gap data to experimental band gap data, particularly in light of the difficulties of predicting the band gap in DFT.

I am not sure of the extent of such work and hesitate to require it.

The manuscript is clearly written and is very readable. I think that the procedure is reasonably well explained and the analysis is sound. The literature in the field is fairly treated. In passing I note that 2D at the top of page 2 should probably be 3D.

To summarise, the manuscript demonstrates successfully a novel approach to predicting materials properties using machine learning and due to the interest, challenge and importance of such methods I recommend it be published in Nature Communications after the authors have considered both extending the set of properties and the comparison to experimental and not only computational data.

Reponses to the reviewers

Reviewer #1, Comment #1:

The paper report a novel graph oriented machine learning technique for analyzing the AFLOWlib repository and predicting mainly band-gaps from the connectivity and a large number of atomic descriptors.

While I like the paper, it is a very clearly written and illustrated attempt at an interesting field, I have difficulties finding arguments for why it should be published in Nat. Comm., I think it belongs in a more specialized journal like J.Chem.Phys.

My major objections are:

1. The choice of studying E-fermi is very questionable. In a PW code this is a parameter set with respect to $G=0$ and carries no physical significance. It says nothing about band alignment as otherwise used as a justification by the authors in the intro and conclusion.

Authors:

We thank the reviewer for the positive feedback.

After some consideration, we agree with the reviewer and the model has been removed entirely from the paper. In its place, by leveraging the same methodology described for the electronic properties, we have constructed models for six thermomechanical properties of real practical use, including: the bulk and shear moduli, Debye temperature, heat capacity (both at constant pressure and volume), and thermal expansion coefficient. The manuscript has been drastically extended.

Here, we discuss additions specific to the six new models:

- Added text in the Methods section (Training Set) describing the new thermomechanical training data.
- Modified the Modeling Work-Flow section to include the new thermomechanical models (includes updated Figure 2), and introduced the public, open-access web application leveraging our models for real time predictions.
- Added a new subsection in Model Interpretation discussing the most significant descriptors to the new thermomechanical models. This includes the addition of new Figure 6 (partial dependence plots).
- Enhanced the Discussion section to highlight the need and practicality for rapid prediction of these properties.

Now the appropriate part of the manuscript reads (in blue):

II. Methods

Data preparation. Two independent datasets were prepared for the creation and validation of the ML models. The training set includes electronic [4, 36–40] and thermomechanical properties [33, 34] for a broad diversity of compounds already characterized in the AFLOW database. This set is used to build and analyze the ML models, one model per property. The constructed thermomechanical models are then employed to make predictions of previously uncharacterized compounds in the AFLOW database. Based on these predictions and consideration of computational cost, several compounds are selected to validate the models' predictive power. These compounds and their newly computed properties define the test set. The compounds used in both datasets are specified in the Supplementary Information.

Training set. Band gap energy data for 49,934 materials were extracted from the AFLOW repository [4, 36–40], representing approximately 60% of the known stoichiometric inorganic crystalline materials listed in the Inorganic Crystal Structure Database (ICSD) [2, 3]. While these band gap energies are

generally underestimated with respect to experimental values [41], DFT+ U is robust enough to differentiate between metallic (no E_{BG}) and insulating ($E_{\text{BG}} > 0$) systems [42]. Additionally, errors in band gap energy prediction are typically systematic. Therefore, the band gap energy values can be corrected ad-hoc with fitting schemes [43, 44]. Prior to model development, both ICSD and AFLOW data were curated: duplicate entries, erroneous structures, and ill-converged calculations were corrected or removed. Noble gases crystals are not considered. The final dataset consists of 26,674 unique materials (12,862 with no E_{BG} and 13,812 with $E_{\text{BG}} > 0$), covering the seven lattice systems, 230 space groups, and 83 elements (H-Pu, excluding noble gases, Fr, Ra, Np, At, and Po). All referenced DFT calculations were performed with the Generalized Gradient Approximation (GGA) PBE [45] exchange-correlation functional and projector-augmented wavefunction (PAW) potentials [46, 47] according to the AFLOW Standard for High-Throughput (HT) Computing [42]. The Standard ensures reproducibility of the data, as well as provides visibility and reasoning for any parameters set in the calculation, such as accuracy thresholds, calculation pathways, and mesh dimensions.

Thermomechanical properties data for just over 3,000 materials were extracted from the AFLOW repository [34]. These properties include the bulk modulus, shear modulus, Debye temperature, heat capacity at constant pressure, heat capacity at constant volume, and thermal expansion coefficient, and were calculated using the AEL-AGL integrated framework [33, 34]. The AEL (AFLOW Elasticity Library) method [34] applies a set of independent normal and shear strains to the structure, and then fits the calculated stress tensors to obtain the elastic constants [48]. These can then be used to calculate the elastic moduli in the Voigt and Reuss approximations, as well as the Voigt-Reuss-Hill (VRH) averages which are the values of the bulk and shear moduli modeled in this work. The AGL (AFLOW GIBBS Library) method [33] fits the energies from a set of isotropically compressed and expanded volumes of a structure to a quasiharmonic Debye-Grüneisen model [49] to obtain thermomechanical properties, including the bulk modulus, Debye temperature, heat capacity, and thermal expansion coefficient. AGL has been combined with AEL in a single workflow, so that it can utilize the Poisson ratios obtained from AEL to improve the accuracy of the thermal properties predictions [34]. After a similar curation of ill-converged calculations, the final dataset consists of 2,829 materials. It covers the seven lattice systems, includes unary, binary, and ternary compounds, and spans broad ranges of each thermomechanical property, including high thermal conductivity systems such as C, BN, BC₅, CN₂, MnB₂, and SiC, as well as low thermal conductivity systems such as Bi₄, Hg₃₃(Rb,K)₃, Cs₆Hg₄₀, BiI₃, and Li.

...

Integrated modeling work-flow. Eight predictive models are developed in this work, including: a binary classification model that predicts if a material is a metal or an insulator and seven regression models that predict: the band gap energy (E_{BG}) for insulators, bulk modulus (B_{VRH}), shear modulus (G_{VRH}), Debye temperature (θ_{D}), heat capacity at constant pressure (C_{p}), heat capacity at constant volume (C_{v}), and thermal expansion coefficient (α_{v}).

Figure 2 shows the overall application work-flow. A novel candidate material is first classified as a metal or an insulator. If the material is classified as an insulator, E_{BG} is predicted, while classification as a metal implies that the material has no E_{BG} . The six thermomechanical properties are then predicted independent of the material’s metal/insulator classification. The integrated modeling work-flow has been implemented as a web application at <http://www.aflow.org/aflow-ml>, requiring only the atomic species and positions as input for predictions.

While all three models were trained independently, the accuracy of the E_{BG} regression model is inherently dependent on the accuracy of the metal/insulator classification model in this work-flow. However, the high accuracy of the metal/insulator classification model suggests this not to be a practical concern.

FIG 2. **Outline of the modeling work-flow.** ML models are represented by orange diamonds. Target properties predicted by these models are highlighted in green.

III. Results

...

Model interpretation. Model interpretation is of paramount importance in any ML study. The significance of each descriptor is determined in order to gain insight into structural features that impact molecular properties of interest. Interpretability is a strong advantage of decision tree methods, particularly with the GBDT approach. One can quantify the predictive power of a specific descriptor by analyzing the reduction of the RMSE at each node of the tree.

Partial dependence plots offer yet another opportunity for GBDT model interpretation. Similar to the descriptor significance analysis, partial dependence resolves the effect of a variable (descriptor) on a property, but only after marginalizing over all other explanatory variables [82]. The effect is quantified by the change of that property as relevant descriptors are varied. The plots themselves highlight the most important interactions among relevant descriptors as well as between properties and their corresponding descriptors.

While only the most important descriptors are highlighted and discussed, an exhaustive list of relevant descriptors and their relative contributions can be found in the Supplementary Information.

For the metal/insulator classification model, the descriptor significance analysis shows that two descriptors have the highest importance (equally), namely $\text{avg}(\Delta H_{\text{fusion}} \lambda^{-1})$ and $\text{avg}(V_{\text{mol}} r_{\text{cov}}^{-1})$. $\text{avg}(\Delta H_{\text{fusion}} \lambda^{-1})$ is the ratio between the fusion enthalpy (ΔH_{fusion}) and the thermal conductivity (λ) averaged over all atoms in the material, and $\text{avg}(V_{\text{mol}} r_{\text{cov}}^{-1})$ is the ratio between the molar volume (V_{mol}) and the covalent radius (r_{cov}) averaged over all atoms in the material. Both descriptors are simple node-specific features. The presence of these two prominent descriptors accounts for the high accuracy of the classification model.

Figure 4 shows the projection of the full dataset onto the dual-descriptor space of $\text{avg}(\Delta H_{\text{fusion}} \lambda^{-1})$ and $\text{avg}(V_{\text{mol}} r_{\text{cov}}^{-1})$. In this 2D space, metals and insulators are substantially partitioned. To further resolve this separation, the plot is split into four quadrants (see dashed lines) with an origin approximately at

$$\text{avg}(V_{\text{mol}} r_{\text{cov}}^{-1}) = 11, \quad \text{avg}(\Delta H_{\text{fusion}} \lambda^{-1}) = 2.$$

Insulators are predominately located in quadrant I. There are several clusters (one large and several small) parallel to the x -axis. Metals occupy a compact square block in quadrant III within intervals 5 <

$\text{avg}(V_{\text{mol}}r_{\text{cov}}^{-1}) < 12$ and $0.02 < \text{avg}(\Delta H_{\text{fusion}}\lambda^{-1}) < 2$. Quadrant II is mostly empty with a few materials scattered about the origin. In the remaining quadrant (IV), materials have mixed character.

Analysis of the projection shown in Figure 4 suggests a simple heuristic rule: all materials within quadrant I are classified as insulators ($E_{\text{BG}} > 0$), and all materials outside of this quadrant are metals. Remarkably, this unsupervised projection approach achieves a very high classification accuracy of 86% for the entire dataset of 26,674 materials. The model misclassifies only 3,621 materials: 2,414 are incorrectly predicted as insulators and 1,207 are incorrectly predicted as metals. This example illustrates how careful model analysis of the most significant descriptors can yield simple heuristic rules for materials design.

The regression model for the band gap energy (E_{BG}) is more complex. There are a number of descriptors in the model with comparable contributions, and thus, all individual contributions are small. This is expected as a number of conditions can affect E_{BG} . The most important are $\text{avg}(\chi Z_{\text{eff}}^{-1})$ and $\text{avg}(C\lambda_{-1})$ with significance scores of 0.075 and 0.071, respectively, where χ is the electronegativity, Z_{eff} is the effective nuclear charge, C is the specific heat capacity, and λ is the thermal conductivity of each atom.

Figure 5 shows partial dependence plots focusing on $\text{avg}(\Delta IP_{\text{bond}})$ as an example. It is derived from edge fragments of bonded atoms ($l = 1$) and defined as an absolute difference in ionization potentials averaged over the material. In other words, it is a measure of bond polarity, similar to electronegativity. Figure 5a shows a steady monotonic increase in ΔE_{BG} for larger values of $\text{avg}(\Delta IP_{\text{bond}})$. The effect is small, but captures an expected physical principle: polar inorganic materials (*e.g.*, oxides, fluorides) tend to have larger E_{BG} .

Given the number of significant interactions involved with this phenomenon, tailoring E_{BG} involves the optimization of a highly non-convex, multidimensional object. Figure 5b illustrates a 2D slice of this object as $\text{avg}(\Delta IP_{\text{bond}})$ and $\text{std}(\Delta IP_{\text{bond}})$ vary simultaneously. Like $\text{avg}(\Delta IP_{\text{bond}})$, $\text{std}(\Delta IP_{\text{bond}})$ is the standard deviation of the set of absolute differences in IP among all bonded atoms. In the context of these two variables, E_{BG} responds to deviations in ΔIP_{bond} among the set of bonded atoms, but remains constant across shifts in $\text{avg}(\Delta IP_{\text{bond}})$. This suggests an opportunity to tune E_{BG} by considering another composition that varies the deviations among bond polarities. Alternatively, a desired E_{BG} can be maintained by considering another composition that preserves the deviations among bond polarities, even as the overall average shifts. Similarly, Figure 5c shows the partial dependence on both the density (ρ) and $\text{avg}(\Delta IP_{\text{bond}})$. Contrary to the previous trend, larger $\text{avg}(\Delta IP_{\text{bond}})$ values correlate with smaller E_{BG} , particularly for low density structures. Materials with higher density and lower $\text{avg}(\Delta IP_{\text{bond}})$ tend to have higher E_{BG} . Considering the elevated response (compared to Figure 5b), the inverse correlation of E_{BG} with the average bond polarity in the context of density suggests an even more effective means of tuning E_{BG} .

A descriptor analysis of the thermomechanical property models revealed the importance of one descriptor in particular, the volume per atom of the crystal. This conclusion certainly resonates with the nature of these properties, as they generally correlate with bond strength [34]. Figure 5d exemplifies such a relationship, which shows the partial dependence plot of the bulk modulus (B_{VRH}) on the volume per atom. Tightly bound atoms are generally indicative of stronger bonds. As the interatomic distance increases, properties like B_{VRH} generally diminish.

Two of the more interesting dependence plots are also shown in Figure 5 (e-f), both of which offer opportunities for tuning the Debye temperature (θ_{D}). Figure 5e illustrates the interactions among two descriptors, the absolute difference in electron affinities among bonded atoms averaged over the material ($\text{avg}(\Delta EA_{\text{atom}})$), and the standard deviation of the set of ratios of the enthalpies of vaporization (ΔH_{vapor}) and atomization (ΔH_{atom}) for all atoms in the material ($\text{std}(\Delta H_{\text{vapor}}\Delta H_{\text{atom}}^{-1})$). Within atom these dimensions, two distinct regions emerge of increasing/decreasing θ_{D} separated by a sharp division at about $\text{avg}(\Delta EA_{\text{atom}}) = 3$. Within these partitions, there are clusters of maximum gradient in θ_{D} —peaks within the left partition and troughs within the right. The peaks and troughs alternate with varying ($\text{std}(\Delta H_{\text{vapor}}\Delta H_{\text{atom}}^{-1})$). Although ($\text{std}(\Delta H_{\text{vapor}}\Delta H_{\text{atom}}^{-1})$) is not an immediately intuitive atom descriptor, the alternating clusters may be a manifestation of the periodic nature of ΔH_{vapor} and ΔH_{atom} [83]. As for the partitions themselves, the extremes of $\text{avg}(\Delta EA_{\text{atom}})$ characterize covalent and ionic materials, as bonded

atoms with similar EA are likely to share electrons, while those with varying EA prefer to donate/accept electrons. Considering that EA is also periodic, various opportunities for carefully tuning θ_D should be available.

Finally, Figure 5f shows the partial dependence of θ_D on the lattice parameters b and c . It resolves two notable correlations: (i) uniformly increasing the cell size of the system decreases θ_D , but (ii) elongating the cell ($c/b \gg 1$) increases it. Again, (i) can be attributed to the inverse relationship between volume per atom and bond strength, but does little to address (ii). Nevertheless, the connection between elongated, or layered, systems and the Debye temperature is certainly not surprising—anisotropy can be leveraged to enhance phonon-related interactions associated with thermal conductivity [84] and superconductivity [85–87]. While the domain of interest is quite narrow, the impact is substantial, particularly in comparison to that shown in Figure 5e.

FIG. 5. Partial dependence plots of the E_{BG} , B_{VRH} , and θ_D models. (a) Partial dependence of E_{BG} on the $\text{avg}(\Delta IP_{\text{bond}})$ descriptor. For E_{BG} , the 2D interaction between $\text{std}(\Delta IP_{\text{bond}})$ and $\text{avg}(\Delta IP_{\text{bond}})$ and between ρ (density) and $\text{avg}(\Delta IP_{\text{bond}})$ are illustrated in panels (b) and (c), respectively. (d) Partial dependence of the B_{VRH} on the crystal volume per atom descriptor. For θ_D , the 2D interaction between $\text{avg}(\Delta EA_{\text{atom}})$ and $\text{std}(\Delta H_{\text{vapor}} - \Delta H_{\text{atom}}^{-1})$ and between crystal lattice parameters b and c are illustrated in panels (e) and (f), respectively.

IV. Discussion

Traditional trial-and-error approaches have proven ineffective in discovering practical materials. Computational models developed with ML techniques may provide a truly rational approach to materials design. Typical high-throughput DFT screenings involve exhaustive calculations of all materials in the database, often without consideration of previously calculated results. Even at high-throughput rates, an average DFT calculation of a medium size structure (about 50 atoms per unit cell) takes about 1,170 CPU- hours of calculations or about 37 hours on a 32-CPU cores node. However, in many cases, the desired range of values for the target property is known. For instance, the optimal band gap energy and thermal conductivity for optoelectronic applications will depend on the power and voltage conditions of the device [88, 93]. Such cases offer an opportunity to leverage previous results and savvy ML models, such as those developed in this work, for rapid pre-screening of potential materials. Researchers can quickly narrow the list of candidate materials and avoid many extraneous DFT calculations—saving money, time, and computational resources. This approach takes full advantage of previously calculated

results, continuously accelerating materials discovery. With prediction rates of about 0.1 seconds per material, the same 32-CPU cores node can screen over 28 million material candidates per day with this frame- work.

Furthermore, interaction diagrams as depicted in Figure 5 offer a pathway to design materials that meet certain constraints and requirements. For example, substantial differences in thermal expansion coefficients among the materials used in high-power, high-frequency optoelectronic applications leads to bending and cracking of the structure during the growth process [88, 93]. Not only would this work-flow facilitate the search for semiconductors with large band gap energies, high Debye temperatures (thermal conductivity), but also materials with similar thermal expansion coefficients.

Reviewer #1, Comment #2:

2. As the authors mention, there are now a number of ML studies of structure property relationships. I don't see how the present really stands out. Especially as no new predictions about materials are made or surprising insights are offered. It is of methodological interest and belongs in a methodological journal.

Authors:

This point serves as one of the primary motivations for our revisions and resubmission. We leveraged our models to make rapid predictions, and subsequently validated them for 770 previously uncharacterized materials. Our investigation achieves an unparalleled affirmation of the validity of machine learning methods in materials science.

To address this concern in the text directly, we:

- Added a subsection to the Methods section (Test Set) regarding the introduction of a new, independent test set to validate the thermomechanical models.
- Added a new section in the Results (Model Performance) presenting the results of this analysis. This includes the addition of (new) Figure 7 and relevant Table II demonstrating these results.

Now the appropriate part of the manuscript reads (in blue):

II. Methods

...

Test set. While nearly all ICSD compounds are characterized electronically within the AFLOW database, most have not been characterized thermomechanically due to the added computational cost. This presented an opportunity to validate the ML models. Of the remaining compounds, several were prioritized for immediate characterization via the AEL-AGL integrated framework [33, 34]. In particular, focus was placed on systems predicted to have a large bulk modulus, as this property is expected to scale well with the other aforementioned thermomechanical properties [33, 34]. The set also includes various other small cell, high symmetry systems expected to span the full applicability domains of the models. This effort resulted in the characterization of 770 additional compounds.

III. Results

Model validation. While the expected performances of the ML models can be projected through five-fold cross validation, there is no substitute for validation against an independent dataset. The ML models for the thermo- mechanical properties are leveraged to make predictions for materials previously uncharacterized, and subsequently validated these predictions via the AEL-AGL integrated framework [33, 34]. Figure 6 illustrates the models' performance on the set of 770 additional materials, with relevant statistical characteristics displayed in Table II.

FIG. 6. Model performance evaluation for the six ML models predicting thermomechanical properties of 770 newly characterized materials. Predicted vs. calculated values for the regression ML models: **(a)** bulk modulus (B_{VRH}), **(b)** shear modulus (G_{VRH}), **(c)** Debye temperature (θ_{D}), **(d)** heat capacity at constant pressure (C_{P}), **(e)** heat capacity at constant volume (C_{V}), and **(f)** thermal expansion coefficient (α_{V}).

property	RMSE	MAE	r^2
B_{VRH}	21.13 (GPa)	12.00 (GPa)	0.93
G_{VRH}	18.94 (GPa)	13.31 (GPa)	0.90
θ_{D}	64.04 (K)	42.92 (K)	0.93
C_{P}	1.47 (k_{B} /cell)	0.92 (k_{B} /cell)	0.98
C_{V}	1.45 (k_{B} /cell)	0.86 (k_{B} /cell)	0.98
α_{V}	1.95×10^{-5} (K^{-1})	5.77×10^{-6} (K^{-1})	0.76

TABLE II. Statistical characteristics of the new predictions for the six thermomechanical regression models (Figure 7).

Reviewer #1, Comment #3:

3. The average predictive quality might be high, but large failures of the model are numerous. This would strongly limit a screening based on the model. Again it is disappointing that no predictions are made and validated.

Authors:

Certainly no model is perfect, but the outliers we observe in what is now the Model Generalizability - Model Validation sections are significant (revealing). In particular, extended analysis of the outliers reveals many to be theoretical/high-pressure structures, offering yet another valuable application for machine learning methods in materials science, large database curation. Additionally, the validation of the six new models to predict thermomechanical properties was particularly fruitful, as the training set was small (<3,000 systems) due to the complexity/cost of such characterizations. The success of these models given such a small training set further corroborates the effectiveness of our unique machine learning approach, as well as its applicability to other complex, but essential, characterizations where training data is limited.

To address this concern in the text directly, we:

- Added a (new) section (Model Generalizability) in which we analyze and discuss how well each of the eight models is expected to perform when given an independent test set.
- Modified Figure 3 to include an analysis of the thermomechanical models. This includes the addition of relevant Table I, which includes relevant statistical information of the five-fold cross-validation analysis for the thermomechanical properties.
- Added a discussion in the Model Performance section detailing outliers in the prediction of the test set.
- Added a discussion of the overall outlier analysis in the Discussion section.

Now the appropriate part of the manuscript reads (in blue):

III. Results

Model generalizability. One technique for assessing model quality is five-fold cross validation, which gauges how well the model is expected to generalize to an independent dataset. For each model, the scheme involves randomly partitioning the set into five groups and predicting the value of each material in one subset while training the model on the other four subsets. Hence, each subset has the opportunity to play the role of the “test set”. Furthermore, any observed deviations in the predictions are addressed. For further analysis, all predicted and calculated results are available in Supplemental Information.

The accuracy of the metal vs. insulator classifier is reported as the area under the curve (AUC) of the receiver operating characteristic (ROC) plot (Figure 3a). The ROC curve illustrates the model’s ability to differentiate between metallic and insulating input materials. It plots the prediction rate for insulators (correctly vs. incorrectly predicted) throughout the full spectrum of possible prediction thresholds. An area of 1.0 represents a perfect test, while an area of 0.5 characterizes a random guess (the dashed line). The model shows excellent external predictive power with the AUC at 0.98, an insulator-prediction success rate (sensitivity) of 0.95, a metal-prediction success rate (specificity) of 0.92, and an overall classification rate (CCR) of 0.93. For the complete set of 26,674 materials, this corresponds to 2,103 misclassified materials, including 1,359 misclassified metals and 744 misclassified insulators. Evidently, the model exhibits positive bias toward predicting insulators, where bias refers to whether a ML model tends to over- or under-estimate the predicted property. This low false-metal rate is fortunate as the model is unlikely to misclassify a novel, potentially interesting semiconductor as a metal. Overall, the metal classification model is robust enough to handle the full complexity of the periodic table.

The results of the five-fold cross validation analysis for the band gap energy (E_{BG}) regression model are plotted in Figure 3b. Additionally, a statistical profile of these predictions, along with that of the six thermomechanical regression models, is provided in Table I, which includes metrics such as the root-

mean-square error (RMSE), mean absolute error (MAE), and coefficient of determination r^2 . Similar to the classification model, the E_{BG} model exhibits a positive predictive bias. The biggest errors come from materials with narrow band gaps, c.f. scatter in the lower left corner in Figure 3b. These materials predominantly include complex fluorides and nitrides. $\text{N}_2\text{H}_6\text{Cl}_2$ (ICSD #23145) exhibits the worst prediction accuracy with signed error $\text{SE} = 3.78$ (eV). The most underestimated materials are HCN (ICSD #76419) and $\text{N}_2\text{H}_6\text{Cl}_2$ (ICSD #240903) with $\text{SE} -2.67$ and -3.19 (eV) [76]. This is not surprising considering that all three are molecular crystals. Such systems are anomalies in the ICSD, and fit better in other databases, such as the Cambridge Structural Database [77]. Overall, 10,762 materials are predicted within 25% accuracy of calculated values, whereas 824 systems have errors over 1 (eV).

Figures 3c-h and Table I showcase the results of the five-fold cross validation analysis for the six thermomechanical regression models. For both bulk (B_{VRH}) and shear (G_{VRH}) moduli, over 85% of materials are predicted within 20 (GPa) of their calculated values. The remaining models also demonstrate high accuracy, with at least 90% of the full training set ($> 2,546$ systems) predicted to within 25% of the calculated values. Significant outliers in predictions of the bulk modulus include graphite (ICSD #187640, $\text{SE} = 100$ (GPa), likely due to extreme anisotropy) and two theoretical high-pressure boron nitrides (ICSD #162873 and #162874, under-predicted by over 110 (GPa)) [78]. Other theoretical systems are ill-predicted throughout the six properties, including ZN (ICSD #161885), CN_2 (ICSD #247676), C_3N_4 (ICSD #151782), and CH (ICSD #187642) [79–81]. Predictions for the G_{VRH} , Debye temperature (θ_{D}), and thermal expansion coefficient (α_{V}) tend to be slightly underestimated, particularly for higher calculated values. Additionally, mild scattering can be seen for θ_{D} and α_{V} , but not enough to have a significant impact on the error or correlation metrics.

Despite minimal deviations, both RMSE and MAE are within 4% of the ranges covered for each property, and the predictions demonstrate excellent correlation with the calculated properties. Of particular interest are the near perfect correlations observed for the heat capacities with the RMSE at just 2 (k_{B}/cell).

FIG. 3: **Five-fold cross validation plots for the eight ML models predicting electronic and thermomechanical properties.** (a) Receiver operating characteristic (ROC) curve for the classification ML model. (b)-(h) Predicted vs. calculated values for the regression ML models: (b) band gap energy (E_{BG}), (c) bulk modulus (B_{VRH}), (d) shear modulus (G_{VRH}), (e) Debye temperature (θ_D), (f) heat capacity at constant pressure (C_P), (g) heat capacity at constant volume (C_V), and (h) thermal expansion coefficient (α_V).

property	RMSE	MAE	r^2
E_{BG}	0.51 (eV)	0.35 (eV)	0.90
B_{VRH}	14.25 (GPa)	8.68 (GPa)	0.97
G_{VRH}	18.43 (GPa)	10.62 (GPa)	0.88
θ_D	56.97 (K)	35.86 (K)	0.95
C_P	2.31 (k_B /cell)	0.84 (k_B /cell)	0.99
C_V	2.01 (k_B /cell)	0.70 (k_B /cell)	0.99
α_V	1.47×10^{-5} (K) $^{-1}$	5.69×10^{-6} (K) $^{-1}$	0.91

TABLE I. Statistical characteristics of the five-fold cross-validated predictions for the seven regression models (Figure 3).

III. Results

Model validation.

...

Comparing with the results of the generalizability analysis shown in Figure 3 and Table I, the overall errors are consistent with five-fold cross validation. Five out of six models have r^2 of 0.9 or higher. However, the r^2 value for the thermal expansion coefficient (α_v) is lower than forecasted. The presence of scattering suggests the need for a larger training set—as new, much more diverse materials were likely introduced in the test set. This is not surprising considering the number of variables that can affect thermal expansion [88]. Otherwise, the accuracy of these predictions are a testament to the effectiveness of the PLMF representation, which is particularly compelling considering: (i) the limited diversity training dataset (only about 11% as large as that available for predicting the electronic properties), and (ii) the relative size of the test set (over a quarter the size of the training set).

In the case of the bulk modulus (B_{VRH}), 665 systems (86% of test set) are predicted within 25% of calculated values. Only the predictions of four materials, Bi (ICSD #51674), PrN (ICSD #168643), Mg_3Sm (ICSD #104868), and ZrN (ICSD #161885), deviate beyond 100 (GPa) from calculated values. Bi is a high-pressure phase (Bi-III) with a caged, zeolite-like structure [89]. The structures of zirconium nitride (wurtzite phase) and praseodymium nitride (B3 phase) were hypothesized and investigated via DFT calculations [90, 91] and have yet to be observed experimentally.

For the shear modulus (G_{VRH}), 482 materials (63% of the test set) are predicted within 25% of calculated values. Just one system, C_3N_4 (ICSD #151781), deviates beyond 100 (GPa) from its calculated value. The Debye temperature (θ_D) is predicted to within 50 (K) accuracy for 540 systems (70% of the test set). BeF_2 (ICSD #173557), yet another cage (sodalite) structure [92], has the largest errors in three models including θ_D (SE = 423 (K)) and both heat capacities (C_p and C_v). Similar to other ill-predicted structures, this polymorph is theoretical, and has yet to be synthesized.

IV. Discussion

...

While the models themselves demonstrated excellent predictive power with minor deviations, outlier analysis revealed theoretical structures to be among the worst offenders. This is not surprising, as the true stability conditions (e.g., high-pressure/high-temperature) have yet to be determined, if they exist at all. The ICSD estimates that structures for over 7,000 materials (or roughly 4%) come from calculations rather than actual experiment. Such discoveries exemplify yet another application for ML modeling, rapid/robust curation of large datasets.

Reviewer #1, Comment #4:

Smaller point:

The "Fragment construction" subsection is a bit repetitive (compare p2c2)

Authors:

Yes, we agree. We have modified the Methods section to reduce this redundancy.

Now the appropriate part of the manuscript reads (in blue):

II. Methods

Universal Property-Labeled Materials Fragments (PLMF).

...

Figure 1 shows the scheme for constructing PLMFs. Given a crystal structure, the first step is to determine the atomic connectivity within it. In general, atomic connectivity is not a trivial property to determine within materials. Not only must the potential bonding distances among atoms be considered, but also whether the topology of nearby atoms allows for bonding. Therefore, a computational geometry approach is employed to partition the crystal structure (Figure 1a) into atom-centered Voronoi-Dirichlet polyhedra [59–62] (Figure 1b). This partitioning scheme was found to be invaluable in the topological analysis of metal organic frameworks (MOF), molecules, and inorganic crystals [63, 64]. Connectivity between atoms is established by satisfying two criteria: (i) the atoms must share a Voronoi face (perpendicular bisector between neighboring atoms), and (ii) the interatomic distance must be shorter than the sum of the Cordero covalent radii [65] to within a 0.25 Å tolerance. Here, only strong interatomic interactions are modeled, such as covalent, ionic, and metallic bonding, ignoring van der Waals interactions. **Due to the ambiguity within materials, the bond order (single/double/triple bond classification) is not considered. Taken together, the Voronoi centers that share a Voronoi face and are within the sum of their covalent radii form a three-dimensional graph defining the connectivity within the material.**

In the final steps of the PLMF construction, the full graph and corresponding adjacency matrix (Figure 1c) are constructed from the total list of connections. **The adjacency matrix A of a simple graph (material) with n vertices (atoms) is a square matrix ($n \times n$) with entries $a_{ij} = 1$ if atom i is connected to atom j , and $a_{ij} = 0$ otherwise.** This adjacency matrix reflects the global topology for a given system, including interatomic bonds and contacts within the crystal. The full graph is partitioned into smaller sub-graphs, corresponding to individual fragments (Figure 1d). While there are several subgraphs to consider in general, the length l is restricted to a maximum of three, where l is the largest number of consecutive, **non-repetitive** edges in the subgraph. This restriction serves to curb the complexity of the final descriptor vector. In particular, there are two types of fragments. Path fragments are subgraphs of at most $l = 3$ that encode any linear strand of up to four atoms. **Only the shortest paths between atoms are considered.** Circular fragments are subgraphs of $l = 2$ that encode the first shell of nearest neighbor atoms. In this context, circular fragments represent coordination polyhedra, or clusters of atoms with anion/cation centers each surrounded by a set of its respective counter ion. Coordination polyhedra are used extensively in crystallography and mineralogy [66].

Reviewer #2, Comment #1:

The main claim of this paper is that the fragment material descriptors introduced by the authors in combination with machine learning approaches allows the development of universal models capable of accurately predicting electronic properties for inorganic materials.

To the best of my knowledge these claims are novel. They are of course a continuation of earlier studies by the authors and others as cited in the paper itself. Machine learning approaches for predicting materials properties is a very active field at the forefront of research and this paper will most definitely be of interest to other researchers in this field. The ideas promoted in this paper may be expected to influence thinking in this field in two main directions at least: (i) establishing feasibility of ML methods in materials properties predictions and (ii) specific contribution of the materials descriptors employed here including the material fragments and the mix of empirical and computational properties. The authors claim is supported by three separate analyses: (i) classification of metal/non-metal materials (ii) prediction of the Fermi energy relative to DFT (iii) prediction of the band gap relative to DFT. While the proposed method is quite successful in predicting these quantities there are some weaknesses arising from the choice of properties. In particular, the Fermi energy is a computational concept and depends strongly on the choice of model and the band gap is a notoriously difficult problem in DFT and the error relative to experiment can be tens of percent.

Authors:

We appreciate the positive feedback.

To the first point, as stated in response to Reviewer #1 Comment #1, we have removed the Fermi energy model from the paper. Six new models have been added in its place extending the suggestions of the reviewers.

To the second next point, we address this concern in the Methods section, under the new Training Set subsection. The particular band gap energy values we train on were calculated using the DFT+*U* approach (effective Hubbard *U* from Hartree Fock), using parameters listed in Ref. [42] (Calderon *et al.*, Comput. Mater. Sci **108 Part A**, 2015). This correction serves to enhance the accuracy of the calculated band gap energy values within the DFT framework, and generally can differentiate between metals and insulators. However, we strongly emphasize that while this paper focuses on DFT calculations as training input, experimentally observed band gap energy values can be used in its place. The framework does not depend on the source or nature of the training data.

Reviewer #2, Comment #2:

I think that the authors' case would be strengthened considerably by modelling a mechanical parameter, such as the bulk modulus, in addition to the purely electronic properties considered. Another issue would be to compare the predicted band gap data to experimental band gap data, particularly in light of the difficulties of predicting the band gap in DFT.

I am not sure of the extent of such work and hesitate to require it.

Authors:

This suggestion serves as the other primary motivation for resubmission. We agree with and followed through with the reviewer's request regarding modeling (thermo)mechanical properties like the bulk modulus. In fact, we modeled other five additional thermomechanical properties, including shear modulus, Debye temperature, heat capacity (at constant pressure and volume), and thermal expansion coefficient. Please see our response to Reviewer #1 Comment #1 for details.

As per Referee2 request for the band gap energy data, please see the answer to Comment #1. Ultimately, this investigation demonstrates that it is feasible to develop predictive models based on elemental/structural properties, but independent of the source of the training data.

Reviewer #2, Comment #3:

The manuscript is clearly written and is very readable. I think that the procedure is reasonably well explained and the analysis is sound. The literature in the field is fairly treated. In passing I note that 2D at the top of page 2 should probably be 3D.

Authors:

The reviewer is correct, sorry for the typo.

Now the appropriate part of the manuscript reads (in blue):

II. Methods

Universal Property-Labeled Materials Fragments (PLMF).

...

In this representation, fragment descriptors characterize subgraphs of the full 3D molecular network.

Reviewer #2, Comment #4:

To summarise, the manuscript demonstrates successfully a novel approach to predicting materials properties using machine learning and due to the interest, challenge and importance of such methods I recommend it be published in Nature Communications after the authors have considered both extending the set of properties and the comparison to experimental and not only computational data.

Authors:

Thanks for the support. Validation and experimental comparison have been addressed at the beginning of this report. We look forward for a positive outcome of the review process.

END OF REPORT

REVIEWERS' COMMENTS:

Reviewer #1 (Remarks to the Author):

It was a pleasure to see how the authors have improved the manuscript. With a relative small effort I think it deserves publication in Nat. Comm.

I still feel the work falls short on the "general interest" scale. The authors now make and validate predictions on thermomechanical properties for a large number of materials. I accept that the very interested reader could get these values from AFLOWlib or from the (very) fine print in the SI.

However, some discussion of these properties is necessary to get a broad interest in the materials science community which would justify publication in a general interest journal like Nat. Comm.

Are there any extreme values (large or small B and G). Are there any interesting combinations?
E.g. large B small G

Smaller points

The authors mention the very good predictive power for the heat capacities. Isn't this just due to Dulong-Petit? Try giving the the heat capacities as k_B/atom instead of k_B/cell

More than 100 citations for a communication is a lot.

Reviewer #2 (Remarks to the Author):

The authors in their resubmission have responded to all the issues raised by this reviewer. In particular, the choice of physical properties to which the proposed method is applied has been extended significantly. Furthermore, the discussion of the validity and application of the model has also been revised.

Therefore this referee recommends that this work should be accepted for publication.

However, prior to final acceptance the authors should make the following modifications which will improve the quality of their presentation.

(i) With respect to heat capacity all the data should be presented in terms of k_B/atom and not k_B/cell . In the latter case most of the variation is dominated by the number of atoms per cell and we do not obtain relevant information in a transparent manner.

(ii) The authors should consider adding the average values of the training set descriptors to Fig 5 eg Fig 5e makes much sense if the average training value for ΔE_{bond} is approx. 3.2. If it is not, then the authors should explain this difference.

Reponses to the reviewers

Reviewer #1, Comment #1:

It was a pleasure to see how the authors have improved the manuscript. With a relative small effort I think it deserves publication in Nat. Comm.

I still feel the work falls short on the "general interest" scale. The authors now make and validate predictions on thermomechanical properties for a large number of materials. I accept that the very interested reader could get these values from AFLOWlib or from the (very) fine print in the SI.

Authors:

We thank the reviewer for positive feedback.

Yes, the original Supplementary Information was quite difficult to read. We have since reformatted the document in *LaTeX* to enhance both its presentation and utility. The newly formatted document is also significantly shorter than the original.

We have attached the document with this submission.

Reviewer #1, Comment #2:

However, some discussion of these properties is necessary to get a broad interest in the materials science community which would justify publication in a general interest journal like Nat. Comm. Are there any extreme values (large or small B and G). Are there any interesting combinations? E.g. large B small G .

Authors:

Yes there are, many of which have already been included in the description of our training set, but characterized by thermal conductivity. We have added to this discussion.

Now the appropriate part of the manuscript reads (in blue):

II. Methods

Training set. ... After a similar curation of ill-converged calculations, the final dataset consists of 2,829 materials. It covers the seven lattice systems, includes unary, binary, and ternary compounds, and spans broad ranges of each thermomechanical property, including high thermal conductivity systems such as C (ICSD #182729), BN (ICSD #162874), BC_5 (ICSD #166554), CN_2 (ICSD #247678), MnB_2 (ICSD #187733), and SiC (ICSD #164973), as well as low thermal conductivity systems such as $Hg_{33}(Rb,K)_3$ (ICSD #410567 and #410566), Cs_6Hg_{40} (ICSD #240038), $Ca_{16}Hg_{36}$ (ICSD #107690), CrTe (ICSD #181056), and Cs (ICSD #426937). Many of these systems additionally exhibit extreme values of bulk and shear moduli, such as C (high bulk and shear moduli) and Cs (low bulk and shear moduli). Interesting systems such as RuC (ICSD #183169) and NbC (ICSD #189090) with a high bulk modulus ($B_{VRH} = 317.92$ GPa, 263.75 GPa) but low shear modulus ($G_{VRH} = 16.11$ GPa, 31.86 GPa) also populate the set.

Reviewer #1, Comment #3:

The authors mention the very good predictive power for the heat capacities. Isn't this just due to Dulong-Petit? Try giving the heat capacities as k_B/atom instead of k_B/cell .

Authors:

We have created two new models to predict the per atom heat capacities at constant pressure and at constant volume. These were built independently of the per cell variants, and thus all analyses (including Feature Importance in the Supplementary Information) have been updated. Indeed, this representation clearly exhibits the Dulong-Petit law. We now comment on this in the Results section. While the per atom variants replace the per cell ones in the revised manuscript, both representations are independently predicted and displayed in our online application.

Now the appropriate part of the manuscript reads (in blue):

III. Results

Model generalizability. ... Note the tight clustering of points just below 3 (k_B/atom) for the heat capacity at constant volume (C_V). This is due to C_V saturation in accordance with the Dulong-Petit law occurring at or below 300 K for many compounds.

FIG. 3: **Five-fold cross validation plots for the eight ML models predicting electronic and thermomechanical properties.** (a) Receiver operating characteristic (ROC) curve for the classification ML model. (b)-(h) Predicted vs. calculated values for the regression ML models: (b) band gap energy (E_{BG}), (c) bulk modulus (B_{VRH}), (d) shear modulus (G_{VRH}), (e) Debye temperature (θ_{D}), (f) heat capacity at constant pressure (C_{p}), (g) heat capacity at constant volume (C_{v}), and (h) thermal expansion coefficient (α_{v}).

property	RMSE	MAE	r^2
E_{BG}	0.51 (eV)	0.35 (eV)	0.90
B_{VRH}	14.25 (GPa)	8.68 (GPa)	0.97
G_{VRH}	18.43 (GPa)	10.62 (GPa)	0.88
θ_{D}	56.97 (K)	35.86 (K)	0.95
C_{p}	0.09 (k_{B}/atom)	0.05 (k_{B}/atom)	0.95
C_{v}	0.07 (k_{B}/atom)	0.04 (k_{B}/atom)	0.95
α_{v}	$1.47 \times 10^{-5} (\text{K})^{-1}$	$5.69 \times 10^{-6} (\text{K})^{-1}$	0.91

TABLE I. Statistical summary of the *five-fold cross-validated predictions* for the seven regression models (Figure 3).

Model validation. ... BeF₂ (ICSD #173557), yet another cage (sodalite) structure [94], has among the largest errors in three models including θ_{D} (SE = -423 K) and both heat capacities (C_{p} ; SE = $0.65 k_{\text{B}}/\text{atom}$; C_{v} ; SE = $0.61 k_{\text{B}}/\text{atom}$).

FIG. 6. **Model performance evaluation for the six ML models predicting thermomechanical properties of 770 newly characterized materials.** Predicted vs. calculated values for the regression ML models: (a) bulk modulus (B_{VRH}), (b) shear modulus (G_{VRH}), (c) Debye temperature (θ_{D}), (d) heat capacity at constant pressure (C_{p}), (e) heat capacity at constant volume (C_{v}), and (f) thermal expansion coefficient (α_{v}).

property	RMSE	MAE	r^2
B_{VRH}	21.13 (GPa)	12.00 (GPa)	0.93
G_{VRH}	18.94 (GPa)	13.31 (GPa)	0.90
θ_{D}	64.04 (K)	42.92 (K)	0.93
C_{P}	0.10 (k_{B}/atom)	0.06 (k_{B}/atom)	0.92
C_{V}	0.07 (k_{B}/atom)	0.05 (k_{B}/atom)	0.95
α_{V}	$1.95 \times 10^{-5} \text{ (K)}^{-1}$	$5.77 \times 10^{-6} \text{ (K)}^{-1}$	0.76

TABLE II. Statistical summary of the *new predictions* for the six thermomechanical regression models (Figure 6).

Reviewer #1, Comment #4:

More than 100 citations for a communication is a lot.

Authors:

This work discusses the calculation, prediction, and (experimental) validation of a plethora of properties. To balance transparency and efficiency, we provide the minimum references needed to reproduce the results.

Reviewer #2, Comment #1:

The authors in their resubmission have responded to all the issues raised by this reviewer. In particular, the choice of physical properties to which the proposed method is applied has been extended significantly. Furthermore, the discussion of the validity and application of the model has also been revised.

Therefore this referee recommends that this work should be accepted for publication. However, prior to final acceptance the authors should make the following modifications which will improve the quality of their presentation.

(i) With respect to heat capacity all the data should be presented in terms of kB/atom and not kB/cell. In the latter case most of the variation is dominated by the number of atoms per cell and we do not obtain relevant information in a transparent manner.

Authors:

The positive review is very much appreciated.

Indeed, the per atom variants of the heat capacity have been constructed and added to the paper, replacing the per cell ones. See our response to Reviewer #1 (Comment #3).

Reviewer #2, Comment #2:

(ii) The authors should consider adding the average values of the training set descriptors to Fig 5 eg Fig 5e makes much sense if the average training value for ΔEA_{bond} is approx. 3.2. If it is not, then the authors should explain this difference.

Authors:

We have calculated the averages for all descriptors highlighted in the work and added it to the Supplementary Information (see Feature Importance). The average of this particular descriptor ($\text{avg}(\Delta EA_{\text{atom}})$) is 2.63. A distinction should be made between the descriptor and the partial dependence function. The latter describes variations of a property (e.g., θ_D) in the context of (two) descriptors. The partition in the plot is a feature of θ_D in this context, and not simply of $\text{avg}(\Delta EA_{\text{atom}})$. Conversely, the intuition we discuss (covalent vs. ionic materials) only applies to the descriptor and not to the partial dependence function.

We show an example table from the Supplementary Information:

I. Feature Importance

variable	importance	mean
$\text{avg}(V_{\text{molar}} C^{-1})$	0.1512	53.99
volume per atom	0.1439	19.87
$\text{avg}(m_{\text{atom}} C)$	0.0812	23.22
$\text{avg}(m_{\text{atom}} \Delta H_{\text{atom}}^{-1})$	0.0589	0.38
$\text{avg}(\Delta(p_P)_{\text{bond}})$	0.0509	4.18
$\text{avg}(p_P Z_{\text{eff}})$	0.0455	28.66
c	0.0264	5.55
$\text{std}(\Delta H_{\text{vapor}} \Delta H_{\text{atom}}^{-1})$	0.0198	16.20
$\text{avg}(\Delta EA_{\text{bond}})$	0.0193	2.63
b	0.0179	5.32

TABLE V. Most significant features of the Debye temperature regression model. The mean value of each feature across the training set is also provided.

END OF REPORT